# Mechanistic model of radiotherapy-induced lung fibrosis using coupled 3D agent-based and Monte Carlo simulations

Nicolò Cogno [1,2,3], Roman Bauer[4] & Marco Durante [1,2,5✉]

## Abstract

**Background** Mechanistic modelling of normal tissue toxicities is unfolding as an alternative to the phenomenological normal tissue complication probability models. The latter, currently used in the clinics, rely exclusively on limited patient data and neglect spatial dose distribution information. Among the various approaches, agent-based models are appealing as they provide the means to include patient-specific parameters and simulate long-term effects in complex systems. However, Monte Carlo tools remain the state-of-the-art for modelling radiation transport and provide measurements of the delivered dose with unmatched precision.

**Methods** In this work, we develop and characterize a coupled 3D agent-based – Monte Carlo model that mechanistically simulates the onset of the radiation-induced lung fibrosis in an alveolar segment. To the best of our knowledge, this is the first such model.

**Results** Our model replicates extracellular matrix patterns, radiation-induced lung fibrosis severity indexes and functional subunits survivals that show qualitative agreement with experimental studies and are consistent with our past results. Moreover, in accordance with experimental results, higher functional subunits survival and lower radiation-induced lung fibrosis severity indexes are achieved when a 5-fractions treatment is simulated. Finally, the model shows increased sensitivity to more uniform protons dose distributions with respect to more heterogeneous ones from photon irradiation.

**Conclusions** This study lays thus the groundwork for further investigating the effects of different radiotherapeutic treatments on the onset of radiation-induced lung fibrosis via mechanistic modelling.

## Plain language summary

Lung cancer leads to a significant number of deaths each year. Radiotherapy is known to be effective in treating lung cancer. However, it can also damage healthy tissue and this limits the dose that can be delivered to the cancer. To estimate the risk of harming healthy tissues in the lung with radiotherapy, mathematical models can be used. We propose a computer-based model to overcome some of the limitations of existing approaches currently used in the clinic. The model incorporates spatial information about the radiation dose and replicates findings observed in mice and humans on lung scarring caused by radiation. With further testing, our model may allow clinicians to better minimize harm to healthy tissues in patients with lung cancer.

[1] Biophysics Department, GSI Helmholtzzentrum für Schwerionenforschung GmbH, 64291 Darmstadt, Germany. [2] Institute for Condensed Matter Physics, Technische Universität Darmstadt, 64289 Darmstadt, Germany. [3] Department of Radiation Oncology, Massachusetts General Hospital and Harvard Medical School, Boston, MA, USA. [4] Department of Computer Science, University of Surrey, Guildford GU2 7XH, UK. [5] Department of Physics "Ettore Pancini", University Federico II, Naples, Italy. ✉email: m.durante@gsi.de

Precision, efficacy, and non-invasiveness have made radiotherapy (RT) a first-choice treatment option nowadays for a large portion of cancer patients. Nevertheless, the risk of developing radiation-induced pathologies remains substantial, and the number of reported injuries increases concurrently with the number of treatments. Notably, more than 50% of cancer patients are treated with RT[1], and previous studies reported that thorax irradiation led to radiation-induced lung injuries (RILI—namely, pneumonitis and fibrosis) in up to 30% of the cases[2]. These pathological conditions are thought to be triggered mainly by the damaged alveolar epithelium, and the resulting inflammation, if not resolved within a few weeks' time, can lead to lung stiffening and eventually death[3]. Consequently, greater efforts are required to improve our understanding of the mechanisms underlying the paths that link radiation damage and toxicity in normal (i.e., non-tumoural) tissues, with the ultimate goal of widening the therapeutic window.

To this aim, normal tissue complication probability (NTCP) models have been implemented that estimate the probability of developing new pathologies as a function of the delivered dose for a given organ[4–6]. However, NTCP models widely employed in clinical practice restrict the set of inputs to the delivered dose distribution and a few macroscopic organ-specific parameters that fail to enfold a mechanistic description of the pathways involved. Moreover, the lack of spatial information in the dose-volume histograms (DVHs) that encode the dose distributions' data results in incomplete descriptions of the treatment plans and subsequent erroneous shifts of the NTCP curves[7,8].

By combining agent-based (AB) modeling and a Monte Carlo (MC) simulator we propose, to the best of our knowledge, an innovative approach in an attempt to address the aforementioned shortcomings. AB modeling is a powerful and relatively recent set of computational techniques that allows simulations of concurrent and independent entities, namely, the agents[9]. Each agent is initialized at the beginning of a new simulation with a set of rules, encoded into its behaviors, and positioned in an environment that can be sensed by and react to the agent itself. The interactions among different agents and between the surrounding environment and the agents themselves can lead to the emergence of elaborated dynamics and provide a framework to model complex systems, such as biological ones. MC methods, on the other hand, have been extensively used to simulate the interaction of radiation with matter[10] and can provide accurate estimates of, among others, dose depositions in biological structures.

In what follows, we provide insights into the onset of radiation-induced lung fibrosis (RILF) in a coupled 3D AB (ABM)—MC model of an alveolar segment. We updated our existing ABM of RILF[11,12] and implemented the alveolar segment geometry within the TOPAS-nBio MC extension[13]. As a result, our coupled model is now able to provide a mechanistic description of the depletion of the functioning distal airways of the lung as well as 3D spatial information about the delivered dose distributions. More specifically, in this work, we develop the coupled model and provide a comparison between the ABM and the ABM-MC models' results. Moreover, the effects of different radiation qualities, as well as the impact of a few damage-associated parameters on the outcomes, are shown. Finally, we demonstrate how the use of a multi-fractionation scheme affects the model's predictions. Temporal fractionation techniques are widely adopted in the clinics[14–16] due to their ability to spare normal tissues without losing efficacy in tumor cell killing[17]. Our results, in agreement with previous experimental studies[18], show a right shift toward higher doses for the same amount of damage when using 5 fractions with respect to a single fraction.

## Methods

**The agent-based model.** Our 3D ABM of RILF in an alveolar segment was developed using the open-source platform BioDynaMo[19], and its implementation was detailed in our previous works[11,12]. The model replicates a small section of the distal airways in the human lungs, namely, an alveolar segment, where three layers of alveoli are stacked in a cylindrical shape. Each layer includes six alveoli, which are modeled as hollow spheres. The surface of each sphere, in turn, is lined by three main cell types: the mesenchymal cells (fibroblasts and myofibroblasts) in the outer layer, the M1 and M2 in the inner layer, and the epithelial cells (types 1 and 2) in the middle layer. Following radiation-induced damage in the epithelial layer, AEC2s can either become apoptotic or turn into a damaged and eventually senescent state. By secreting multiple chemokines and cytokines[20], the senescent cells can both damage the surrounding healthy AEC2 (where the minimum number of senescent cells needed to damage a healthy one is regulated by the bystander threshold parameter) and trigger an immune response led by M1 and M2. Provided by the capillaries (which are not simulated by our model), M1 and M2 (whose phagocytic fraction and phagocytic index are set with custom parameters) are gathered in the alveoli and remove the senescent cells[21,22]. At the same time, the healthy AEC2 increase their proliferation and differentiation rates in an attempt to replenish both their own population and the type 1 alveolar epithelial cells (AEC1)[23,24]. Ultimately, the disruption of the healthy epithelium and the secretion of growth factors from the type 2 macrophages concurrently stimulate the expansion of the mesenchymal compartment, which, in turn, saturates the alveolus with ECM components. The severity of the damage at later time points (e.g., months/years after the treatment) is measured via the RSI, whose definition (proposed in our previous work[11]) was inspired by the concept of FSU in the critical volume model of NTCP[25,26] and by the FI introduced in the work of Zhou et al.[18]. The RSI reads as follows:

$$\mathrm{RSI} = \sqrt{\overline{\triangle\mathrm{ECM}_{\mathrm{conc}}} \uparrow * \triangle V_{\mathrm{surv,FSU}} \downarrow}\left(g^{1/2}\right) \quad (1)$$

Where the two factors describe the average increase in the concentration of the ECM across the whole simulation space (in g/cm$^3$) and the decrease in the volume of surviving FSUs (in cm$^3$), represented by the alveoli. More specifically, $\triangle V_{\mathrm{surv,FSU}} \downarrow$ is computed as the total segment volume (assuming a spherical shape for the alveoli) times the surviving fraction of the FSUs.

Other equations of note (which were described in more detail in our previous work and were used to fit the simulations' output in the section "Results") include the increase in the average ECM concentration across the whole simulation space for the early and late components (where $\triangle\mathrm{ECM}_{\mathrm{max}}$, $\gamma$ and $D_{50}$ represent the saturation value for the ECM increase, the steepness of the sigmoid and dose at 50% of the $\triangle\mathrm{ECM}_{\mathrm{max}}$, respectively)[27–29], given by the following

$$\triangle ECM(D) = \frac{\triangle ECM_{\mathrm{max}}}{1 + e^{4*\gamma*\left(1 - \frac{D}{D_{50}}\right)}}\left(\frac{g}{\mathrm{cm}^3}\right) \quad (2)$$

The RSI (where $A$, $\gamma$ and $\mathrm{ED}_{50}$ represent the saturation value for the RSI, the steepness of the sigmoid and dose at 50% of the maximum RSI, respectively, and erf is the error function)[18] is expressed as

$$RSI(D) = \frac{1}{2} * A * \left\{1 - erf\left(\sqrt{\pi} * \gamma * \left(1 - \frac{D}{\mathrm{ED}_{50}}\right)\right)\right\}\left(g^{1/2}\right) \quad (3)$$

It's worth noting that Eq. (3) was used in our previous work[11] to fit the late component of the increase in the average ECM

concentration as well. However, given its similarity with Eq. (2), in this work, the latter has been used for both the early and the late components so as to improve the overall readability and have a dedicated equation for the RSI. Finally, the FSU survival probability, derived from the LQ and critical volume NTCP models[25,26] (where $N_{AEC2}$ is the total number of healthy AEC2 in an alveolus in homeostatic conditions), can be written as

$$
\begin{aligned}
P_{\text{surv},FSU}(D) &= 1 - P_{\text{kill},FSU} = 1 - \prod_{i=1}^{N_{\text{AEC2}}} P_{\text{kill},cell} \\
&= 1 - \left(P_{\text{kill},cell}\right)^{N_{\text{AEC2}}} = 1 - \left(1 - e^{-\alpha D - \beta D^2}\right)^{N_{\text{AEC2}}}
\end{aligned}
\tag{4}
$$

With respect to the previous version, the current model brings with it new features and noteworthy changes that will be detailed in what follows. Among them, modeling of the apoptotic AEC2s has been implemented to increase the accuracy of the simulations. Once an AEC2 has changed its state to apoptotic (either due to irradiation damage or aging), free movement is hampered, and the time to removal from the simulation is drawn from a Poisson distribution[30].

It is worth emphasizing that the weakening of the immune system due to radiation-induced damage is not simulated by our model. In fact, to avoid additional complexity and potential uncertainty in the parameters, it is assumed that only the AEC2 cells can be damaged by the radiation, while all the other cell populations, including M1 and M2, are left unaffected. However, by tuning the aforementioned parameters that regulate the phagocytic fraction (i.e., the fraction of macrophages that can phagocyte senescent cells) and index (i.e., the maximum number of senescent cells that can be phagocyted by a macrophage) the damage to the immune system could be easily implemented.

At the beginning of a new simulation, each alveolus is initialized with the three cell compartments described before. Cells (i.e., the agents) are distributed at random positions on the alveolar surface and assigned type-specific behaviors. Among them, cell migration plays an important role and has been optimized in our latest model (see our previous work for additional details[11]). Macrophages and mesenchymal cells travel along spherical arcs in random directions to patrol the alveolar space and maintain homeostatic ECM concentrations, respectively. AEC1 and AEC2, however, are capable of neighborhood-informed migration and thus move preferentially towards depleted zones in order to repopulate them. Finally, damaged and senescent epithelial cells move in random directions at a slower speed than the healthy ones but might not be able to move at all at times. In our updated model, every cell movement is followed by a check on the cell's final position. If the cell does not happen to be located on the spherical surface it belongs to (mainly due to approximation errors when the spherical arc is computed), the cell is translated to the appropriate radial distance while keeping the polar and azimuthal angles fixed.

As in our previous model, the simulations run in a closed cubic space of side 2000 μm that encompasses the whole alveolar segment. A diffusion grid overlaps the simulation space and dissects it into smaller cubic voxels, whose number can be adjusted by the user. The simulated cells can both measure and change the concentration of the substances in the voxel they are located in. Once secreted, substances can both diffuse and decay and, in some cases, be depleted by other substances. Our model simulates, among others, such coupled substances, i.e., matrix metalloproteinases (MMP), tissue inhibitors of metalloproteinases (TIMP), and ECM (interaction mechanisms are outlined in our previous works[11,12]). The system of reaction-diffusion partial differential equations (PDEs) for all the involved substances is then solved using a forward in time—central in space (FCTS)

method with user-defined boundary conditions (BC). In its updated version, both the binding coefficients, the target, and the BC for the substances in the model can be specified in the initialization phase. In particular, we set Neumann BC (as opposed to our previous model) with a constant value to ensure a net zero flux and mimic inter-compartmental communication. Moreover, we increased the duration of one time-step for the solution of the system of PDEs in the diffusion grid to 20 s. By doing so, we could shorten simulation times while still fulfilling both the stability (i.e., Courant–Friedrichs–Lewy) and positivity conditions.

The model simulates the diffusion, decay, and, in some instances, the depletion of ten different substances (their time evolution following the irradiation in a single fraction can be found in Supplementary Fig. 1). These extracellular substances altogether constitute a network that serves as an interface between the cell populations[11,12] which, for instance, senescent AEC2 can use to gather the monocytes and M2 to increase the proliferation of the mesenchymal cells as well as regulate the concentration of the ECM. Although simulating such a number of extracellular substances doesn't come without disadvantages (such as a decrease in the model's performance), it provides a more accurate representation of the mechanisms that underlie the RILF and allows the users to test the outcomes of new therapies by adjusting the model's parameters.

**The Monte Carlo model.** TOPAS-nBio is built on top of the Geant4-DNA[31] toolkit and extends the functionalities of the TOPAS platform[32]. It allows MC simulations at the microscopic and nanoscopic scale while simplifying the modeling of biological structures as well as consistent measurements of radiobiological quantities.

To investigate the effect of realistic dose distributions on our model of the alveolar segment, we reimplemented its structure from scratch using the TOPAS-nBio platform. Our extension consists of 3 new classes that are used to both build the geometry of the cylinder and score the dose delivered to its building blocks.

When a new simulation is performed, an envelope for the alveolar segment is first built as a cubic box with a side of 900 μm. A for loop runs then through a vector containing the centers of the 18 alveoli (whose positions can be either hard coded or provided in a separate file) and smaller containment boxes are generated. The modeling proceeds by instantiating a generic spherical structure with a temporary radius for the cells, and a second loop runs through all the alveoli positions. Within its body, the alveoli are built by placing the cells on three spherical surfaces (as described in the section "The agent-based model"). More specifically, parameters for each cell (i.e., position in 3D, color that indicates the cell type and size) are loaded from external files and, given the inherent repetitive nature of the geometry, the parameterization technique[33] is used to allow for faster rendering and simulation times (see Fig. 1). As opposed to those in the ABM, cells in this model do not have behaviors, are static and differ only by color and size.

Hits within cells are processed using a custom scorer that provides both the total dose deposited within the cells as well as the average dose for each alveolus. By checking the cells' size, non-epithelial cells are filtered out, and the computation of the dose is skipped. For the epithelial cells, however, the total energy deposited by the hitting particles is summed and converted to Gy units. At the end of a simulation, the total dose delivered to each cell is exported to a text file along with a unique identifier and the identifier of the alveolus to which the cell belongs. Moreover, the average dose delivered to (the epithelial cells of) each alveolus is computed and exported together with the alveolus identifier to a

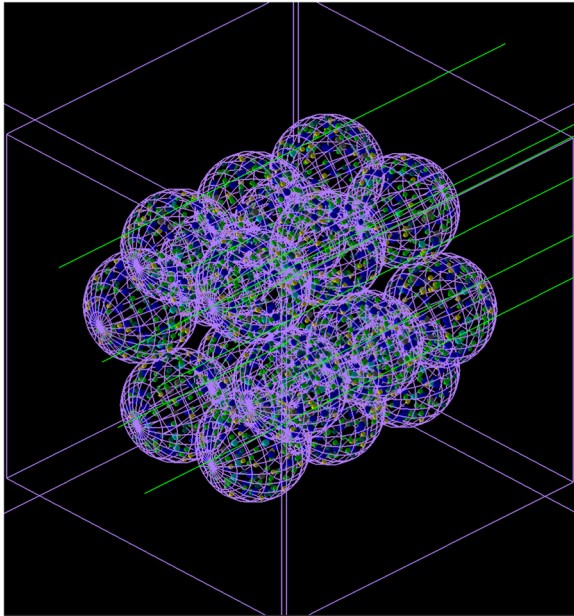 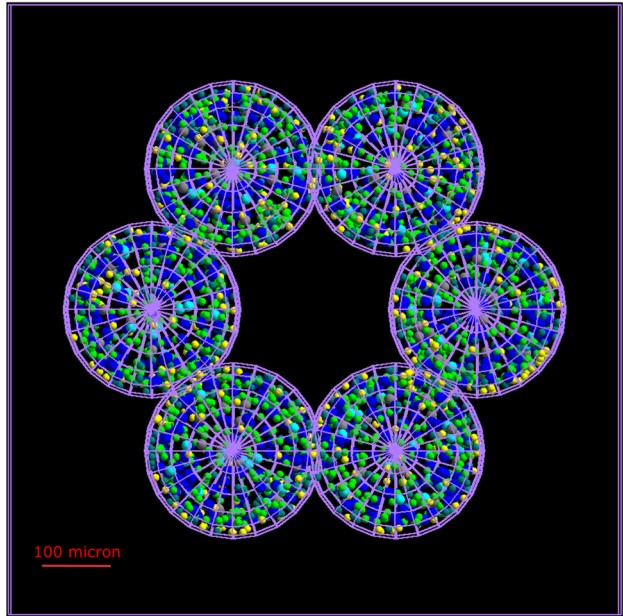

**Fig. 1 Alveolar segment model in TOPAS-nBio.** Alveolar segment model built with TOPAS-nBio and irradiated with 10 keV gamma particles. The 18-alveoli structure is inscribed in a cubic box, while each alveolus has a spherical envelope. Cells have a spherical shape and differ by size and color. The outer space is empty, while cells are filled with water. The side of the outer box is 900 μm long, while the diameter of each alveolus measures 260 μm.

second file. Given that our previous work identified the alveoli as the FSU of the lung, the latter can be useful to, for example, characterize the dose distribution on a bigger scale than the cellular one.

Finally, simulations can be run using the parameter control system as described in work by Schuemann et al.[13]. In the parameter file, the total world volume is defined as a cubic box with the same size as the simulation space of the ABM. The alveolar duct is then placed at its center, and the envelope material is set to vacuum, while we assumed that water is the sole component for the cells. With regard to the particle sources, we tested both an isotropic source located at the center of the alveolar duct and an external beam with one to four coplanar fields. Following the measurements of the average delivered dose per alveolus and outlined in the section "Results", the homogeneity of the dose distribution was maximized by the external beam with four fields. The motion of the beam was implemented by using a step function with variable positions on the XZ plane and 90° rotations about the y-axis, while the cut-off shape was set to "rectangle". The energy and type of the particles to be included in the beam were also set via the parameter file, as well as the angular and energetic spreads, which were set to zero. Settings for our custom scorer were also adjusted via the parameter control system.

**Coupling the models**. The coupled model consists of the aforementioned AB and MC models, each equipped with import/export functionalities that allow data exchange. In addition to this, overall control of the workflow is assigned to a bash script (see Figs. 2 and 3)

The sequence starts with a 1000-step long run of the ABM that does not require any external data to generate the alveolar segment structure in healthy conditions. We tuned our ABM by running 20-day-long simulations in homeostatic conditions and ensuring that both the number of cells and the average concentration of the substances remained constant and matched previously published results (more details on the tuning procedure as well as the parameters can be found in our previous work[11]). To ensure the consistency of the results, 10 independent

experiments are performed for each parameter set, and the output is used to feed the next step. In particular, we implemented a BioDynaMo standalone operation[19] that can be activated using a parameter and is triggered at the end of each simulation. With it, the cells' parameters, together with the position and the diameter of the alveoli and the concentration per voxel of all the simulated substances, can be exported. The data files for the cells contain the cells' position, their radius, an RGB color code that identifies the type and additional type-specific parameters, such as the number of phagocyted cells for the macrophages or the number of steps spent above the bystander damage threshold for the healthy AEC2. Moreover, the identifier of the alveolus each cell belongs to is used to generate a new file for each alveolus.

Subsequently, the control script is used to set the total number of fractions as well as the steps for the ABM, the number of repeated experiments (where a new seed is selected for each run), and the target dose (through the number of histories of the MC model). The MC simulation is then run: TOPAS-nBio builds a new structure for the alveolar segment (using the data files mentioned above and the procedure described in the section "The Monte Carlo model"), irradiates it and generates a summary of the dose deposited in each epithelial cell and the average dose per alveolus. It is worth noting that the duration of the irradiation is orders of magnitude shorter than the average cell's behavior time, and therefore, it is assumed that cells don't change their position and/or state within it.

The workflow proceeds with a new run of the ABM, and custom parameters are used to enable the reading of the external damage files. However, the concentration data are read in subsequent runs only as the duration of the preparatory simulation does not ensure steady-state values for all the substances. Cells' data are read sequentially as well and used to rebuild the alveolar segment structure and assign the type-specific behaviors to the agents. For each epithelial cell, the external damage file from the last step is checked, and if a match is found, the LQ model is used to determine the fate of the cell, which will either remain healthy or become senescent or apoptotic. More specifically, the MC simulator provides the delivered dose per AEC2. The selected number of histories is sufficiently small so

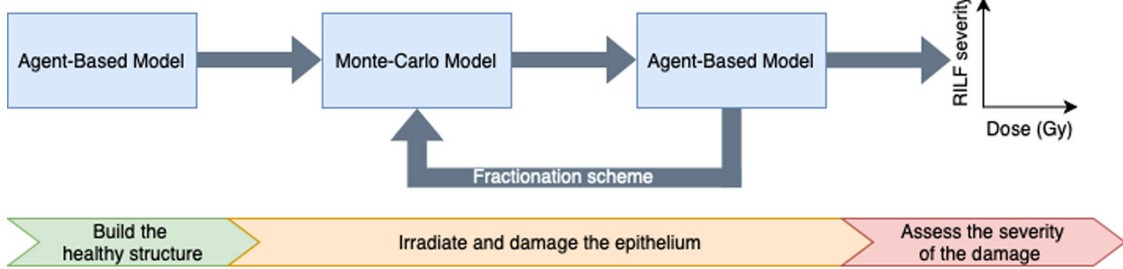

**Fig. 2 Agent-based - Monte Carlo model (ABM-MC) schematic workflow.** Schematic representation of the implemented workflow. The agent-based model (ABM) is used to generate the alveolar segment structure in homeostatic conditions. The data are then fed into the Monte Carlo (MC) model, which replicates the geometry and irradiates it. Information about the dose delivered to each cell is then loaded into the ABM, which simulates the evolution of the system. If a multi-fractionation scheme is set up, the data exported from the ABM are used as input to the MC. If not, the data can be directly analyzed to determine whether radiation-induced lung fibrosis (RILF) onsets and, if so, its severity.

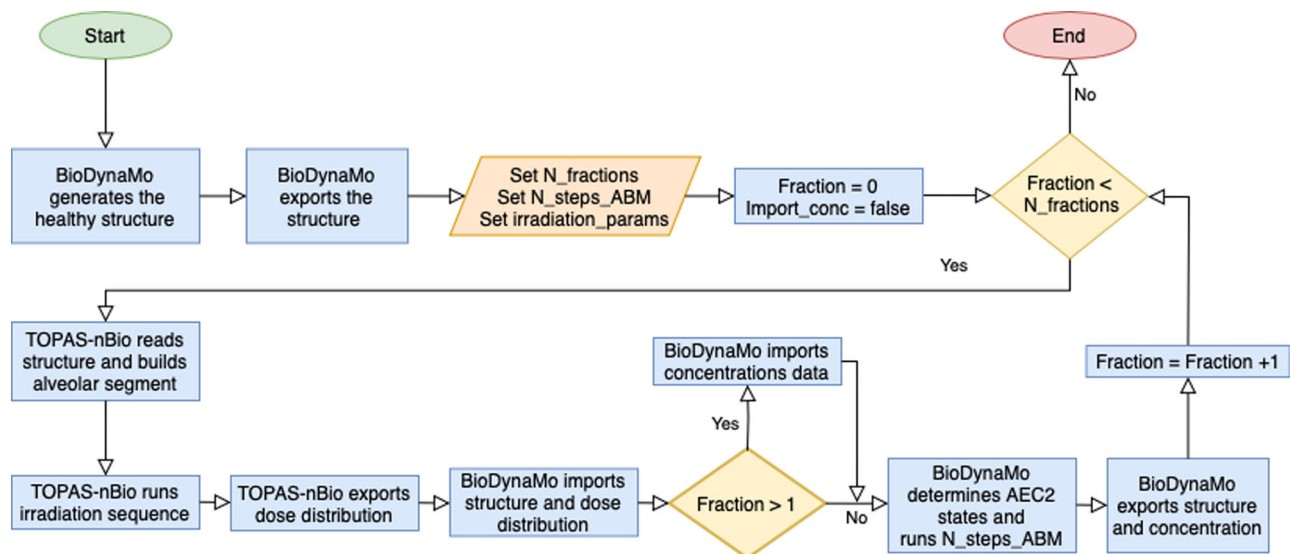

**Fig. 3 Agent-based - Monte Carlo model (ABM-MC) interaction algorithm.** BioDynaMo runs the agent-based model (ABM) for 1000 steps, generates the healthy structure, and exports the alveoli positions and the cells' data. The number of fractions, the number of steps for the ABM, and the irradiation parameters are then set into the control script, which triggers the first run of the Monte Carlo (MC) model using TOPAS-nBio. The structure is loaded from the data exported during the previous step, and the alveolar segment is irradiated. Consequently, the dose distribution is exported, and the data is used as input for the ABM to determine the cells' fate. A new run of the ABM is then performed and the segment structure, together with the cells' state and their position, are exported. If a new fraction is to be delivered, the data are fed into the MC model. Otherwise, the workflow terminates.

that the resulting dose is lower than those simulated but large enough such that the differences between the average doses per alveolus are small. Assuming a constant LET, the delivered dose per cell depends only on the particle fluence and thus grows linearly with the number of histories. Therefore, prior to establishing the damage, the ABM computes the average dose per cell and a scale factor to align it with the desired dose. Subsequently, for each cell, the scaled dose is computed, and a random number, drawn from a uniform distribution, is compared against the AEC2 LQ survival curve value at that dose (whose parameters were derived from previously published experimental data[34], as detailed in our previous works[11,12]). A new simulation is then performed (whose duration is set in the bash script), and depending on the number of fractions, the output is used to feed either a longer run of the ABM or a new irradiation phase via the MC simulator.

In terms of statistical sampling, all the simulations presented throughout this work were performed 10 times for each set of parameters, including the dose, the bystander threshold, the phagocytic fraction, and the index of the macrophages. To ensure independence between different runs, a new seed was selected in the control script at the beginning of each run and used to initialize the MC simulator. This resulted in different damage distributions that, in turn, led to variations among the initial conditions of the ABM. For the ABM, a constant seed was set. However, BioDynaMo compensated by automatically providing random number generators whose seed was linked to the identifier of the thread in use to simulate an agent, thus ensuring independence between and within runs. Each ABM simulation was run for 5.184.000 steps, resulting in a total simulation time of 1200 days (given the 20 s-long time steps). Additionally, 24 h were simulated in the ABM between the delivery of each dose for the multi-fractionation schemes (thus adding 5 simulated days for the delivery of 5 fractions), while single fractions were assumed to be instantaneous.

A notable feature of our framework is its ease of use and customization. Besides BioDynaMo and TOPAS-nBio, users only need an executable file to simulate the ABM and text files that conform to the TOPAS Parameter Control System. As mentioned above, parameters such as the number of fractions, the number of time steps, and dose per fraction, as well as the number of experiments, can be defined in the custom control script. If there

is no need to update the agents' behaviors or the structure of the alveolar segment in the ABM, it's enough to compile the provided C++ code with BioDynaMo to generate the executable file (the healthy structures are included as text files in the Supplementary Software 1). As for the TOPAS-nBio parameters that define, for example, the particle type and energy, the number of histories, and the configuration of the particle source, they do not need to be compiled and are already included in Supplementary Software 1. Once the location of the aforementioned files, the control script can be executed from a shell to perform both the AB and the MC simulations sequentially. The simulations' output is ROOT[35] files that contain the number of cells and the concentration of each substance at multiple time points and locations, stored for further analysis.

Preparatory simulations of the ABM used to generate the healthy structures, as well as those in between fractions, were run on a MacBook Pro laptop with a 2.3 GHz Quad-Core Intel Core i5 processor and 8 GB RAM. For longer (requiring more than 1 h) or parallel simulations, however, a compute node of the Lichtenberg HPC system with $2 \times 2.3$ GHz Intel Cascade-Lake AP 48-cores processor (96 total cores) and 384 GB RAM was used. Besides, the average runtime required to perform 1200-day-long simulations was 3 h. The reduced simulation times (enabled mainly by the optimizations made in the BioDynaMo framework[36] and the longer simulation steps) allowed us to perform longer simulations (with respect to our previous model, up to 1200 days) and improve the statistics by performing more experiments.

**Reporting summary**. Further information on research design is available in the Nature Portfolio Reporting Summary linked to this article.

## Results
### Characterization of the dose distributions in the alveolar segment.
Before running the full ABM-MC model, different setups of the photon source for the irradiation of the alveolar segment were tested, and the corresponding dose distributions were compared. More specifically, doses were delivered using both an external beam with 4 coplanar fields and an isotropic source located at the center of the alveolar segment. Multiple particle energies $E_\gamma$ were tested in a range that included both high cross sections and more realistic/clinically plausible values. The total number of histories (i.e., emitted particles per field) was set to 10 million, which ensured both reasonable simulation times and hits even at the energies associated with the lowest cross sections. Finally, each experiment was performed 10 times, and the results were averaged.

As can be seen from Fig. 4, while the dose delivered with the external beam at $E_\gamma = 1$ keV and $E_\gamma = 10$ keV could be fitted to Gaussian distributions, the dose distribution from the isotropic irradiation and those obtained for higher $E_\gamma$ showed multiple distinct peaks, long right tails and high counts at 0 Gy.

Given the symmetry of the resulting dose distribution and its closer resemblance to the beam setups used in the clinics, the external beam was used in all the experiments performed in this study. Among the energies tested, $E_\gamma = 10$ keV was selected as the standard due to its more realistic homogeneity compared to the distribution obtained with $E_\gamma = 1$ keV. This notwithstanding, the effect of the latter dose distribution on the model's outcomes was also investigated (see the next section).

We further analyzed the effect of an increase in the number of histories in the dose distribution delivered by the external beam. Raising the number of particles from 40 million (for 4 fields) to 120 million did not alter the shape of the dose distribution

substantially but required a much higher computational cost. Therefore, the lower value was set in the parameter file of the MC model and used in the following simulations.

**Simulation outcomes: ABM-MC vs. ABM**. To characterize the coupled ABM-MC model, we compared the main simulation outcomes for $E_\gamma = 1$ keV and $E_\gamma = 10$ keV against those from our previous ABM model[11]. In particular, we evaluated the surviving fraction of the functional subunits (FSUs), the RILF severity index (RSI), and the early and late extracellular matrix (ECM) fluctuations in the ECM concentration as a function of the delivered dose. To ensure that the system reached the steady state (both in terms of cell number and substance concentration, see Supplementary Figs. 1 and 2), we simulated 1200 days. In the ABM-MC model, the dose was delivered once at the beginning of the simulation, and the parameters were set to the same values as those in the ABM model (specifically: phagocytic fraction = 100%, phagocytic index = 1, apoptotic-to-senescent ratio = 0 and bystander threshold = 2).

As can be seen from Fig. 5, both the early and the late components of the ΔECM (i.e., the increase in the average ECM concentration across the whole simulation space for the ABM-MC model) could be fitted with the Eq. (2) and are qualitatively in agreement with experimental data[27–29]. In particular, the ΔECM in Fig. 5a, c exhibits a sigmoidal response to the delivered dose, as shown by Defraene et al.[27,29], with the highest values reached within the first 3–4 months following the irradiation, as in Konkol et al.[28] (see also Supplementary Fig. 1). As expected, the ΔECM increases as a function of the dose and reaches a plateau around 20 Gy for the early component and 25 Gy for the late component. Due to the higher fraction of mesenchymal cells in the inflammatory phase (i.e., before the macrophages are able to clean the senescent cells), the $\Delta ECM_{max}$ of the early phase is more than twice as much as that in the late phase, and the plateau is reached at lower doses as the total amount of ECM that can be stored in the alveolar space is limited.

Figure 5c shows the surviving fraction of the FSUs as a function of the dose. For doses equal to or smaller than 7.5 Gy in the ABM-MC model, the survival is close to 100%, which implies that no alveolus is fully depleted from the healthy type 2 alveolar epithelial cells (AEC2), neither directly or indirectly. Similarly, Citrin et al.[37] observed full recovery in AEC2 cells of mice irradiated at doses ≤ 5 Gy. Moreover, fitting of the dataset with Eq. (4) shows better agreement for both values of $E_\gamma$ than that observed for the results from the ABM model alone, as well as improved consistency at the lower doses.

Finally, the late component ΔECM and the FSU surviving fraction were combined into the RSI, a surrogate measure of the RILF severity, presented in Fig. 5d. Both the datasets obtained from the ABM-MC model could be fitted using Eq. (3) with trends in agreement with experimental findings about the fibrosis index (FI) from Zhou et al.[18] (see the next section for a direct comparison), and, interestingly, close saturation values ($D \sim 25$ Gy for photons in one fraction).

Overall, the outcomes from the ABM-MC model are qualitatively in agreement with experimental results found in the literature[18,27–29], as similar trends were observed. However, a quantitative assessment could not be achieved, mainly because of the nature of the FI and ΔECM, originally expressed in terms of quantities that can't be discerned from our model (such as the functioning volume and the variation in the radiodensity), but also partly due to the microscopic scale of our model. Notably, our previous model showed increased damage (i.e., lower surviving fractions and higher ΔECM) at the same doses for both $E_\gamma$ values, and we argue that the reason behind that lies in

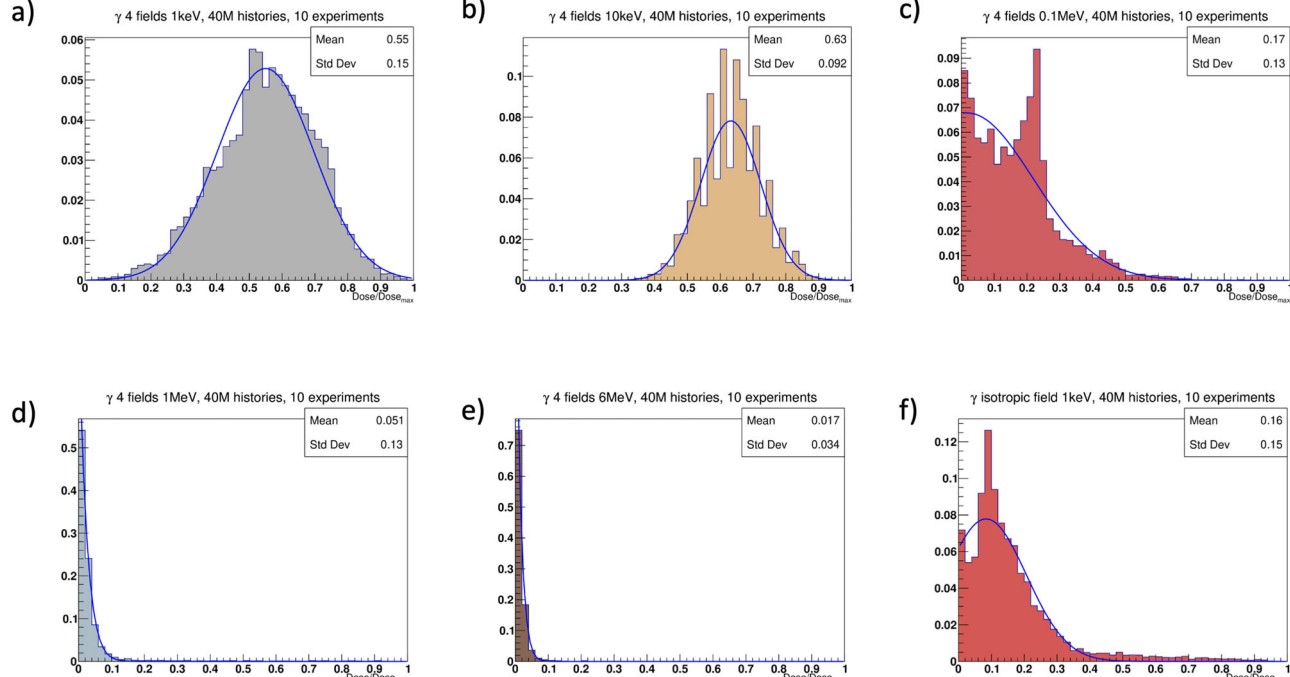

**Fig. 4 Photons dose distribution histograms.** Dose distribution histograms for different setups in the alveolar segment model irradiated with photons. On the x-axis, doses are shown as percentages of the maximum delivered dose, and the blue curves are the best Gaussian fits for the histogram data. Panels **a** through **e** correspond to 4-field external beams with 40 million histories (i.e., particles) for different photon energies. In panel **f**, the dose distribution resulting from an isotropic source (i.e., a point source located at the center of the alveolar duct) is reported.

the heterogeneity of the different dose distributions. In fact, using the dose distribution obtained with the higher $E_\gamma$ (narrower and thus more similar to that used in our older model) resulted in all the dose-response curves shown above being shifted toward the old model's results.

**Temporal fractionation.** To assess the ability of the alveolar segment model to repair the damaged tissue in between irradiations, we implemented a 5-fraction scheme using the ABM-MC model. The temporal fractionation involved the delivery of relatively small doses (see Table 1) 24 h apart, thus allowing the type 1 and 2 macrophages (M1 and M2) to gather (triggered by the secretion of monocyte chemoattractant protein 1, MCP-1, by the damaged AEC2) and start removing the senescent cells. Moreover, the healthy AEC2 could replenish the depleted alveoli. After a 1-day simulation, the system was "frozen," and the structure of the alveolar segment was loaded into TOPAS-nBio for the following irradiation.

The number of fractions and the doses were matched to those used in the experiments presented in the work by Zhou et al.[18], which was used to evaluate our previous ABM model.

Figure 6 presents a comparison of the main outcomes from the ABM-MC model, using 1 and 5 fractions as a function of the total delivered dose. The results were fitted using Eqs. (3) and (4).

As expected, a right shift in both the late component ΔECM and in the FSU survival was observed. Consequently, an isoeffect was measured at higher total doses for the RSI when using 5 fractions with respect to 1 fraction. Similar RSI values for the two fractionation schemes were measured for total doses equal to or lower than 10 Gy and above 35 Gy due to the high variability in the FSU survival at low doses and the formulation of the RSI, which forces it to saturate at high doses. Of note, the FSU survival data for both the single and 5 fractions were accurately described by the linear quadratic (LQ)[38]—critical volume model (i.e., Eq. (4)), even at high doses.

The results for the early and late ΔECM from our ABM-MC model with five fractions not only could be fitted using the equations provided by previous experimental studies by Defraene et al.[27,29] and Konkol et al.[28] for the dose-density increase response but also matched the observations from Zhou et al.[18] about the right shift of the RSI (a surrogate of the severity of the RILF, as is the FI) when the number of fractions increased from 1 to 5. Although the study of Zhou et al.[18] was performed on mice and the RSI can exclusively be regarded as an approximation of the FI, Fig. 6c shows that $\gamma$ and $ED_{50}$ for the two indexes (rescaled to the same saturation value) are consistent. Concerning the RSI and FI for 5 fractions, an agreement was not observed for $ED_{50}$, but the fit curves showed a comparable steepness, as confirmed by the $\gamma$ values and close saturation doses. In a different study by Citrin et al.[37], mice irradiated at 17.5 Gy, 5 × 5 Gy, and 5 × 6 Gy developed lung fibrosis and exhibited poor survival. These findings strongly correlated with a low AEC2 survival and high senescence, similar to what we observed.

**Targeted response assessment.** Although performing a full sensitivity analysis on the ABM-MC model would be intractable due to the high number of parameters (see Supplementary Data 1) and long runtimes, a targeted one aimed at key parameters can provide insights into the model's function.

The majority of the model's parameters (excluding those that govern the reaction-diffusion equations and those that determine the geometry of the model, found in the literature[11,12]) are related to the cells' turnover and secretion rates. These, in particular, were tuned in our first ABM[12], which identified the parameter that controls the influx of macrophages as the most influential among those analyzed. Other damage-associated parameters were introduced in our second ABM[11], which modeled mechanistically the onset of RILI. In that study, we determined the optimal values for the phagocytic fraction and phagocytic index, i.e., 100% and 1. Moreover, we showed how

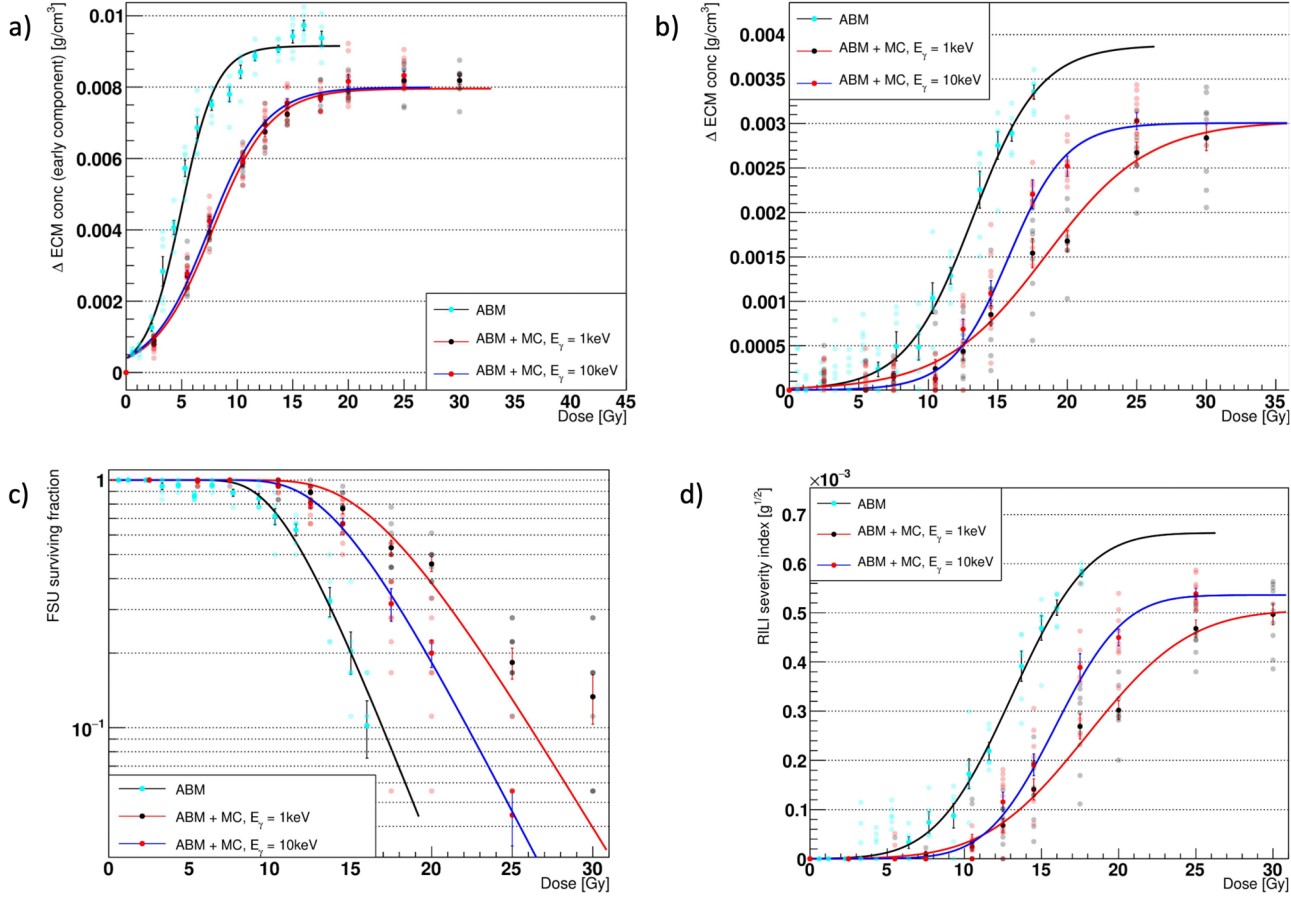

**Fig. 5 Agent-based model (ABM) vs. ABM - Monte Carlo (ABM-MC) outcomes comparison.** Comparison of the major outcomes from the agent-based model (ABM) (cyan markers, black fit curves) and the ABM - Monte Carlo (MC) model using photons in one fraction with $E_\gamma = 1\,keV$ (black markers, red fit curves) and $E_\gamma = 10\,keV$ (red markers, blue fit curves). Panels **a** and **b** show the increase in the average extracellular matrix (ECM) concentration across the whole simulation space after 90 days (early component) and 1200 days (late component). In panel **c**, the surviving fraction of the alveoli (i.e., the functional subunits (FSUs)) in logarithmic scale at the end of the simulation is reported. In panels **a**–**d**, the error bars represent the standard error of the mean (SEM) for $n = 10$ (ABM + MC model) and $n = 6$ (ABM model) independent experiments. The late $\Delta$ECM and the FSU survival are combined to provide the radiation-induced lung fibrosis severity index (RSI), outlined in panel **d**, where the error bars were obtained by propagating the error from the FSU survival and the $\Delta$ECM increase measurements.

| Table 1 Fractionation scheme doses. | | | | | | | | | |
|---|---|---|---|---|---|---|---|---|---|
| Dose per fraction (Gy) | 0.5 | 1.1 | 1.5 | 2 | 4 | 5 | 6 | 7 | 8.5 |
| Total dose (Gy) | 2.5 | 5.5 | 7.5 | 10 | 20 | 25 | 30 | 35 | 42.5 |
| Doses per fraction and corresponding total doses delivered in 5 fractions to gauge the ability of the model to simulate normal tissue-sparing effects. | | | | | | | | | |

different values of the apoptotic-to-senescent ratio affected the survival of the FSUs. We assessed the best value for the bystander threshold as well (i.e., 2), but given that the results from the ABM-MC model exhibited decreased damage for the same doses with respect to our previous model, we evaluated the impact of a faster spread of the indirect damage (by setting the bystander threshold to 1) to determine if the effects could offset each other. The bystander threshold dictates the minimum number of senescent AEC2 necessary to damage a healthy AEC2 that is located in their neighborhood. It is, therefore, the main regulator for controlling the speed of the spread of indirect damage. In fact, the probability that a healthy AEC2 will be indirectly damaged depends on the time spent in the neighborhood of senescent cells, and this, in turn, increases only if the number of senescent neighbors exceeds the bystander threshold (as introduced in the work by McMahon et al.[39]).

As can be seen from Fig. 7, even though the ABM-MC model showed reduced damage with respect to the ABM alone, it could not compensate for the increased spread of the indirect damage. In fact, lowering the bystander threshold led to FSU survivals <100% and $\Delta$ECM > 0, even at very low doses. As a consequence, the RSI never dropped to 0, and the FSU survival showed poor agreement with Eq. (4). Therefore, the impact on the overall model's predictions was substantial.

To further investigate the sensitivity of our model to the damage-associated parameters, we altered the rate at which radiation-damaged cells turn into senescent as well as their radiosensitivity. Specifically, to see if lowering these parameters could compensate for the effect of the bystander threshold highlighted above, we halved the damaged-to-senescent rate and reduced the $\alpha$ and $\beta$ parameters of the LQ model for the cell survival by 10%. The amount of change for the LQ parameters

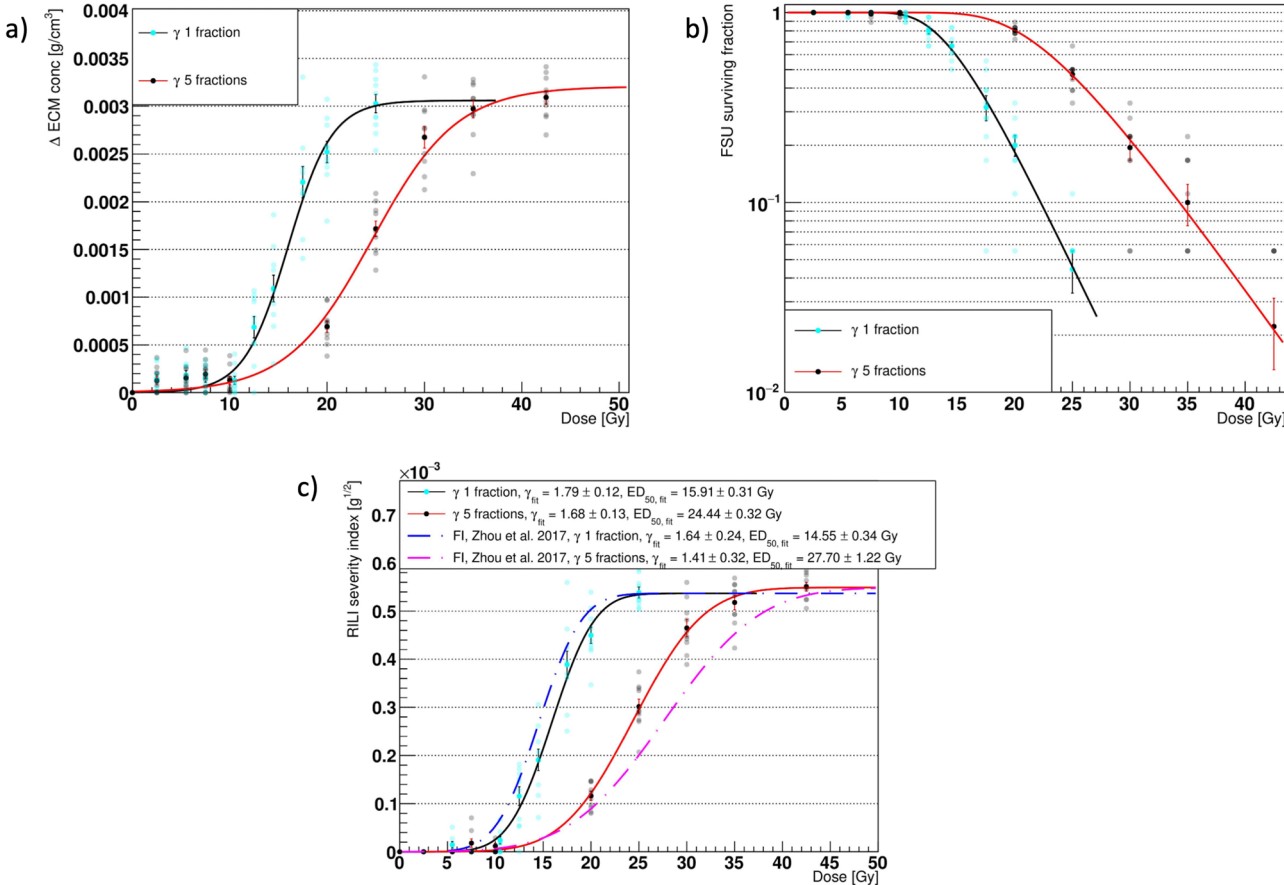

**Fig. 6 Model outcomes comparison for different fractionation schemes.** Comparison of the major outcomes from the agent-based - Monte Carlo model (ABM-MC) using photons in 1 fraction (black markers, red fit curves) and 5 fractions (red markers, green fit curves). Panels **a**–**c** show the increase in the average Extracellular Matrix (ECM) concentration (late component), the surviving fraction of the functional subunits (FSU), and the radiation-induced lung fibrosis severity index (RSI), respectively, at the end of the simulation. The error bars in panels **a** and **b** are the Standard Error of the Mean (SEM) for $n = 10$ in dependent experiments, while those in panel **c** were obtained by propagating the errors in the RSI definition. In panel **c**, the Fibrosis Index (FI) from Zhou et al.[18] for photons in 1 fraction (blue dashed curve) and 5 fractions (fuchsia dashed curve) rescaled to the RSI at saturation is plotted for comparison.

was chosen so as to remain within the error bars computed in our previous work[11], while that of the damaged-to-senescent rate, intrinsically uncertain, was reduced as much as the bystander threshold. As expected, these changes affected the model's outcome by shifting all the curves to higher doses, as shown in Fig. 7. In fact, the lower LQ parameters resulted in fewer cells being damaged by the irradiation, while a smaller damaged-to-senescent rate provided the M1 and M2 with more time to remove the senescent cells, resulting in an enhanced damage mitigation. However, the extent of the shift was marginal and could not balance the impact of the bystander threshold, highlighting the low sensitivity of the model to small variations in the aforementioned parameters.

**Radiation qualities**. The use of charged particles in the clinical setting has seen rapid spread in recent times due to the increased conformality of the dose deposition to the target volume with respect to the photons[40]. Moreover, experimental studies[41] reported differences in the dynamics of the normal tissue response between lung cancer patients treated with photons and protons. We used TOPAS-nBio, coupled to our ABM model, to compare the effects of different radiation qualities on the alveolar segment. In particular, we simulated the irradiation of the lung structure with 2 million protons delivered with an external beam using 4 coplanar fields in one fraction.

TOPAS-nBio can generate standard DNA damage files for each cell that is hit by one or more particles, thus providing punctual information about each and every interaction between the particle and the DNA. However, the process on a common laptop is relatively slow, and the amount of data that is stored for each cell can be very large, thus making it prohibitive for thousands of cells as those in the alveolar segment. Therefore, our simulations did not take into account the differences in the patterns generated by photons and protons when damaging the DNA of the AEC2 cells. Different simulation outcomes are thus solely the result of different dose distributions.

We simulated 60 MeV protons as previous studies have reported negligible differences in the DNA damage response between proton and photon irradiations in the plateau region for this energy[42] and very low linear energy transfer (LET, <2.5 keV/μm) at short distances (<5 mm)[43]. This requirement ensured very small LET variations along the proton tracks (as the diameter of the alveolar segment is <1 mm) and allowed us to use the LQ parameters derived from the survival curve of the AEC2 irradiated with photons.

Figure 8 shows the dose distribution for the protons expressed as a fraction of the maximum delivered dose. When compared with the photon histograms in Fig. 4, the distribution of the proton exhibits a more pronounced and narrower peak, and the data points align well with a Gaussian distribution.

When comparing the main simulation outcomes from the different radiation qualities, a noticeable shift towards lower doses

**Fig. 7 Model outcomes comparison for different parameter values.** Comparison of the major outcomes from the agent-based - Monte Carlo model (ABM-MC) using photons in 1 fraction with bystander threshold = 2 (i.e., standard conditions, cyan markers, black fit curve), bystander threshold = 1 (black markers, red fit curves), halved damaged-to-senescent rate (red markers, blue fit curves) and 10% lower radiosensitivity (blue markers, cyan curves). Panels **a–c** show the increase in the average extracellular matrix (ECM) concentration (late component), the surviving fraction of the functional subunits (FSUs), and the radiation-induced lung fibrosis severity index (RSI), respectively, at the end of the simulation. The error bars in panels **a** and **b** are the standard error of the mean (SEM) for $n = 10$ independent experiments, while those in panel **c** were obtained by propagating the errors in the RSI definition.

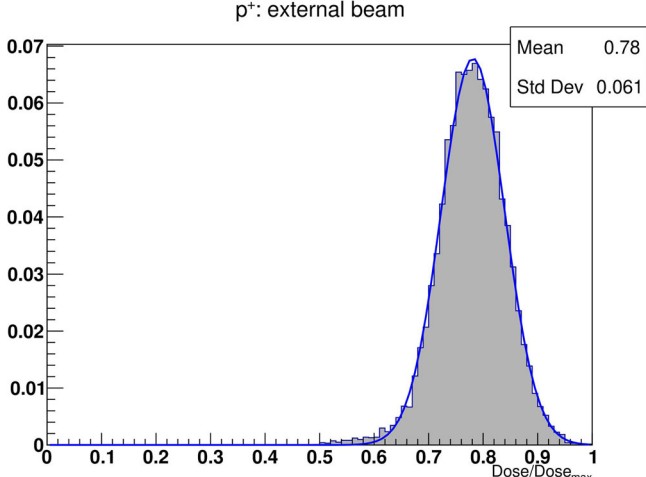

**Fig. 8 Dose distribution histogram for the alveolar segment model irradiated with protons.** The dose was delivered using 4 fields from an external beam and 2 million histories in total. On the x-axis, doses are shown as percentages of the maximum delivered dose, and the green curve is the best Gaussian fit for the histogram data.

for the same effect (i.e., same ΔECM and FSU surviving fraction), shown in Fig. 9, was observed for the protons. This, in turn, resulted in higher RSI at the same dose, similar to what was found when comparing different fractionation schemes for the photons (see Fig. 6), although of a smaller extent.

Given that the FSU survivals could be fitted by the LQ—critical volume model from Eq. (4) for both the photons and the protons, we introduced a relative biological effectiveness[44] relative to the FSU survival ($RBE_{FSU}$). The $RBE_{FSU}$ was defined as the ratio of the absorbed dose from the photons to the absorbed dose from the protons, resulting in the same effect, and provides a quantitative measure of the relative effectiveness of the different radiations in depleting the FSUs. The definition reads as follows:

$$RBE_{FSU} = \frac{D_\gamma}{D_{p^+}}\bigg|_{sf\,\gamma = sf\,p^+} \tag{5}$$

As can be seen from Fig. 9b, small variations were observed for the $RBE_{FSU}$ at the 3 isoeffects considered. While it is generally assumed an RBE = 1.1 for protons relative to photons at the same cell survivals[45], with lower RBEs for higher reference doses, we measured slightly increasing values: 1.12, 1.12, and 1.15 for 50%, 37%, and 10% survival, respectively. Although these values closely match the clinical assumptions, it's worth noting that, as shown

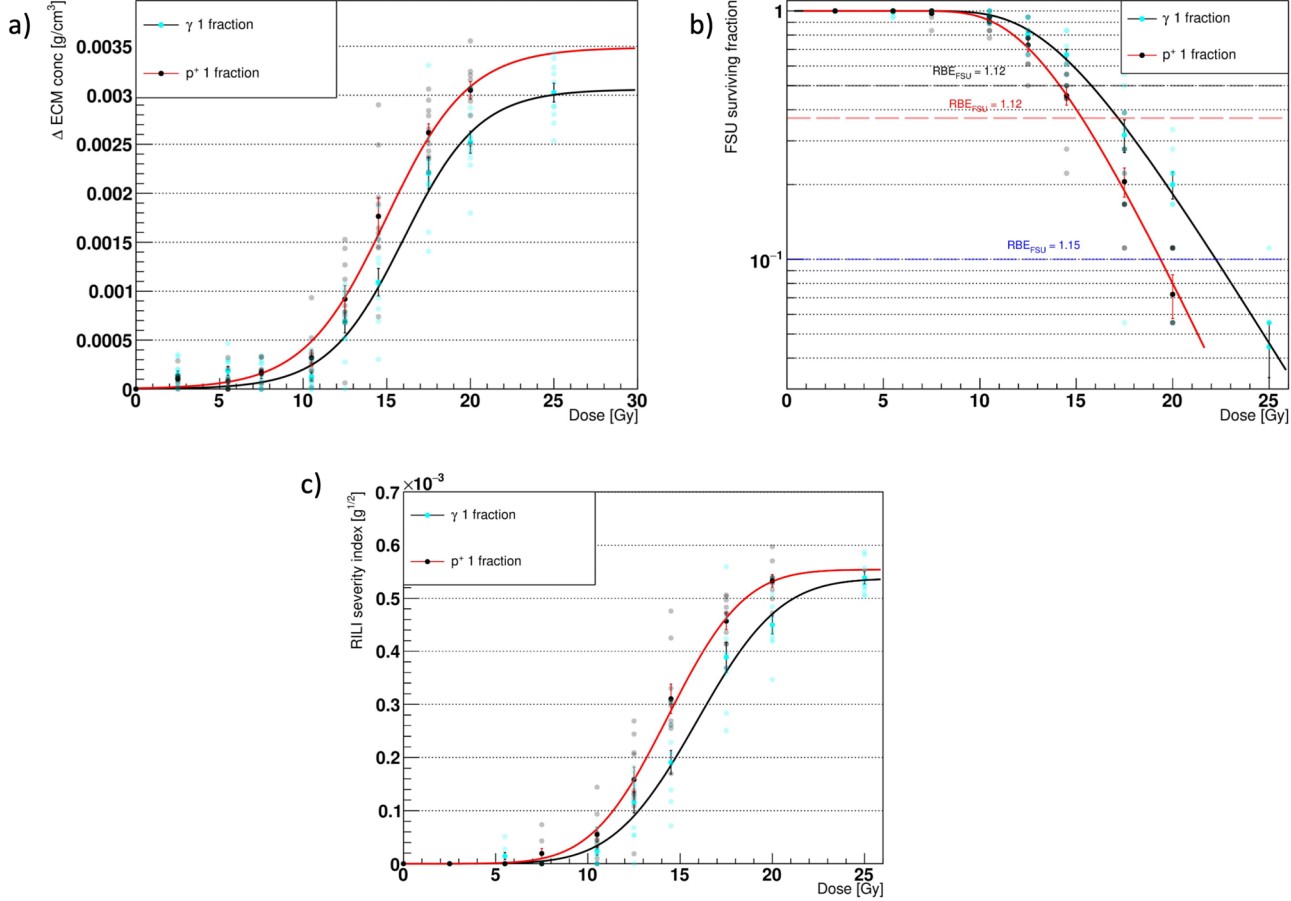

**Fig. 9 Model outcomes comparison for different radiation types.** Comparison of the major outcomes from the agent-based - Monte Carlo model (ABM-MC) using protons (black markers, red fit curves) and photons (cyan markers, black fit curves), both in 1 fraction. Panels **a**–**c** show the increase in the average extracellular matrix (ECM) concentration (late component), the surviving fraction of the functional subunits (FSUs), and the radiation-induced lung fibrosis severity index (RSI), respectively, at the end of the simulation. Panel **b** also provides the relative biological effectiveness (RBE) values for the surviving fraction of the FSUs (RBE_{FSU}) at 50%, 37%, and 10% survival. The error bars in panels **a** and **b** are the standard error of the mean (SEM) for $n = 10$ independent experiments, while those in panel **c** were obtained by propagating the errors in the RSI definition.

previously in Fig. 5, our model is strongly affected by the broadness of the dose distributions used. Therefore, the differences observed in Fig. 9b might just reflect heterogeneities in the dose distribution at a scale that would hold little relevance in a clinical setting. Moreover, it should be emphasized that the RBE_{FSU} is not an RBE in the traditional sense, as it's defined in terms of the survival of the FSUs fitted with a LQ—critical volume model (given by Eq. (4)). As a side note, using the universal survival curve[46] and the linear-quadric-linear[47] formalism to model the $1 - P_{kill, cell} = P_{surv, cell}$ in Eq. (4) resulted in almost identical FSU survival curves and thus no difference in the resulting RBE_{FSU} (see Supplementary Fig. 3).

## Discussion
Despite being a crucial component in a large variety of cancer treatments, the efficacy of RT is intrinsically hindered by normal tissue toxicities[48]. Increasingly sophisticated NTCP models[7] have been implemented to estimate risk probabilities from patients' DVH, but the lack of mechanistic information constrains superficial representations of the underlying mechanisms and prevents patients-specific parameters from being taken into account.

To address the aforementioned shortcomings, we have previously implemented an ABM that simulates the onset of RILF in an alveolar segment[11]. In this work, we outlined the development

of a coupled ABM-MC model, where the alveolar segment structure was rebuilt using TOPAS-nBio[13] and linked to an updated version of our previous model, with a custom interface to handle the communication between the simulation engines. As discussed in our former work, the ability of the model to replicate experimental results was assessed via dose–response curves for the FSU (i.e., the alveoli) survival, the ECM increase, and the RSI, while the output datasets were fitted using Eqs. (3) and (4).

More specifically, in the new implementation, the ABM is used to generate the initial structure in healthy conditions, and afterward, the cells' position, type, and size are exported. Using this data as input, the MC tool builds the structure in real time, and irradiation is simulated based on the information provided via the parameter control system. Accordingly, realistic dose distributions can be simulated and dose depositions at the cell scale registered. The generated data are then sent back into the ABM, where the absorbed doses are used to determine the fate of each cell using the LQ model. Subsequently, the ABM model is run, and the onset of RILF is simulated. Finally, to replicate temporal fractionation schemes, the workflow can be executed multiple times by setting the number of fractions in the control script.

We compared different setups of the photons' source and energy $(E_\gamma)$ by characterizing the resulting relative dose distributions in the alveolar segment. Our results indicated that combining an external source with 4 coplanar fields and $E_\gamma = 10$ keV maximized the dose homogeneity. Therefore, this

configuration was selected as the default for the following simulations.

In accordance with the results from our previous model, the outcomes of the ABM-MC approach (for a single fraction) qualitatively matched published experimental results. In particular, the ΔECM showed two distinct components, an early one that peaked at around 3 months from the irradiation and a late one as the model reached the steady state, as outlined by Konkol et al.[28] and Bernchou et al.[49]. Both the datasets exhibited a sigmoidal response as the dose increased and could be fitted using Eq. (2), as observed by Defraene et al.[27,29]. The surviving fraction of the FSUs was fitted using Eq. (4) and showed good agreement with the LQ–LQ-critical volume model at all doses. Finally, Eq. (3) accurately described the RSI, which mimics the FI introduced by Zhou et al.[18] and provides a surrogate measure of the RILF. Of note, despite showing consistency with the previous ABM model, it can be seen from Fig. 5 that all the curves were shifted towards higher doses for the same outcomes. We attributed these differences to the heterogeneity of the dose distributions and proved our assumptions by showing that the broader dose distribution obtained with 1 keV photons resulted in even more shifted curves. It is worth noting that in the old model, the delivered dose was used to determine the initial damaged fraction, which was the same for all the cells in an alveolus. Conversely, the ABM-MC simulates an actual dose distribution using the TOPAS-nBio module, and each cell, given the absorbed dose, has a certain probability of being damaged in accordance with the LQ model. This notwithstanding, the high sensitivity to the shape of the dose distribution shown by our model urges readers to approach these findings with a degree of skepticism.

Temporal fractionation schemes as a way to spare the normal tissue as sublethal damages are repaired and depleted areas repopulated are widely employed in clinics[14–16]. Accordingly, we compared the results from 1-fraction with those from 5-fractions using the ABM-MC model. As expected, the outcomes in Fig. 6 show a shift towards higher total doses for the alveolar segment irradiated with 5 fractions and could be fitted using Eqs. (3) and (4) for the ΔECM and RSI. Notably, the transition of the RSI to higher total doses from 1 to 5 fractions mimics the findings by Zhou et al.[18] for the photons, although the experiments were performed on mice. Quantitative agreement between the RSI and the FI was observed, as the scaled fit curves showed comparable values for $\gamma$ (for both 1 and 5 fractions) and $ED_{50}$ (for 1 fraction). The model is thus able to replicate normal tissue sparing as the macrophages clean the senescent cells and prevent excessive spread of the damage via bystander mechanisms while the AEC2 repopulates the depleted alveoli. It is worth noting that the actual DNA damage is not taken into account by our model, and therefore, the repair of sublethal damages was not simulated. However, it was recently shown by Liberal et al.[50] that residual DSBs in normal tissue cells exhibit low predictive power for cell survival, and thus our assumptions remain valid.

A targeted response assessment performed on the model as an alternative to a computationally intractable full sensitivity analysis revealed insights into the model's function. Given that the ABM-MC model showed lower damage for the same dose with respect to our previous ABM alone, we investigated the effect of an increase in the speed of the damage spread on the model outcomes by lowering the bystander threshold. Moreover, we reduced the cells' radiosensitivity and damaged-to-senescent rate to see whether these parameters could shift the model's predictions toward higher doses to the same extent as the bystander threshold. As shown by Fig. 7, the reduced damage of the ABM-MC could not compensate for the increase in the bystander effect, which affected all the response curves substantially and prevented full recovery even at the lowest doses. Conversely, lowering the radiosensitivity and damaged-to-senescent rate had a marginal effect on the results, assessing the model's resilience to their changes.

To further characterize the ABM-MC model, we simulated a monoenergetic 60 MeV proton irradiation using the same setup for the source as that previously described for the photons. Differences with respect to the photon irradiation were found in the resulting dose distribution (see Fig. 8), which exhibited a narrower peak at 80% of the maximum delivered dose. As a result, despite the DNA damage and repair mechanisms were not taken into account, the model outcomes differed substantially from those obtained when the alveolar segment was irradiated with photons. Figure 9 shows increased ΔECM and lower FSU survival for the same doses when protons were used, resulting in a higher RSI and similar to what Zhou et al.[51] observed when mice were irradiated using high LET particles. Furthermore, we compared the ability of the different radiation qualities to deplete the alveoli from the AEC2 by computing the RBE at different FSU survivals (that thus differs from the traditional RBE), a metric that accounts for long-term effects as it was computed when the model reached the steady state and the alveolar segment was thus allowed to recover. Given that for the organs with parallel architecture, such as the lungs, the FSU survival plays a key role when estimating the probability of radiation-induced toxicity, future studies could benefit from the $RBE_{FSU}$ as a tool for the comparison of different radiation qualities on the NTCP. Although we measured an average $RBE_{FSU} = 1.13$, considerably close to the value used in clinical practice, it's crucial to acknowledge that the use of 10 keV photons has resulted in heterogeneous dose distributions. This, in turn, could be the underlying reason for the RBE effects observed in this study, which may not be seen in real exposures.

Overall, these findings capture the unique feature of the model (and, more generally, the added value that ABMs can bring to the field of radiation oncology) to be susceptible not only to variations in the average values (i.e., different average doses, as equation-based models would be) but also to local changes (i.e., different dose distributions). This feature, in turn, could be exploited to overcome the lack of sensitivity of the DVH parameters to spatial heterogeneities. Moreover, the influence of changes at multiple scales (such as in the tissue architecture or number of cells of a certain type) on the long-term evolution of the disease can be incorporated and simulated without the need to pause the model. For instance, the key role played by the bystander effect in the onset of RILF has been emphasized by multiple studies[37,52,53], and earlier works highlighted that when radiation-induced lung toxicities are modeled mechanistically, inflammation-induced tissue damage should be taken into account[54]. Such radiation-triggered paracrine senescence could not be integrated into the traditional NTCP models. Additionally, in the same model, both normal tissue and cancer responses to irradiation could be simulated, together with, for instance, the effect of an immunotherapeutic treatment performed at different points in time[55]. Furthermore, our model could be promptly adapted to include subcellular networks, possibly modeled with ODEs, to simulate more accurately the cell dynamics, such as DNA damage and repair mechanisms or senescence-associated pathways. Conversely, at a larger scale, the whole model could be used as a single agent in a multi-agent system to simulate radiation-induced damages at the tissue level. If the local interaction dynamics implemented in this work were extended to such a higher scale, per-voxel dose distributions from clinical treatment plans could be used as an input, and the results compared against those of the current ABM-MC. As shown previously, our ABM-MC predicted slightly different outcomes when the AEC2 radiosensitivity was lowered by 10%. Analogously, the model

could be used to simulate the impact of interpatient variability in radiosensitivities or heterogeneous radiosensitivity distributions on a treatment outcome. Further, given that our model simulates the diffusion of multiple substances, the effect of spatial heterogeneities in the substance's concentration (such as oxygen, which plays an important role in cell survival[56]) due to different treatments or concurrent conditions could be investigated.

In conclusion, this work could serve as a framework to investigate the effect of different fractionation schemes and dose distributions on the severity of RILF while taking into account individualized parameters in the perspective of advancing precision medicine in RT treatments. At the same time, it provides a preliminary validation of the proposed approach, highlighting the potential of ABM combined with MC to inform radiobiological studies. Future developments might concentrate on taking into account the DNA damage and repair mechanisms in order to estimate the fractions of healthy, apoptotic, and senescent cells more accurately.

## Data availability

Source Data for the main figures in the manuscript with statistical analyses are provided in Supplementary Data files 2–4. Raw simulation results are available from the corresponding author on reasonable request.

## Code availability

The code used to implement the ABM-MC model can be found in Supplementary Software 1 and is also available at https://doi.org/10.5281/zenodo.1018563[57]. Open source software used for the ABM model: BioDynaMo, version 1.04, available at https://github.com/BioDynaMo/biodynamo. Open source software used for the MC model: TOPAS-nBio, version 2.0, available at https://github.com/topas-nbio/TOPAS-nBio.

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

## Acknowledgements

Calculations for this research were conducted on the Lichtenberg high-performance computer of the TU Darmstadt. The authors would like to thank the BioDynaMo consortium and attendants of the work meetings for helpful comments and suggestions.

## Author contributions

Conceptualization: N.C.; methodology: N.C.; software: N.C.; writing—original draft preparation: N.C.; writing—review and editing: M.D. and R.B.; visualization: N.C.; supervision: M.D. and R.B. All authors have read and agreed to the published version of the paper.

## Funding

## Competing interests

The authors declare no competing interests.
