## [Peer Review File · Communications Medicine]

Reviewers' comments:

Reviewer #1 (Remarks to the Author):

The work detailed in the manuscript "Mechanistic model of radiotherapy induced lung fibrosis using coupled 3D Agent-Based and Monte Carlo simulations" by Cogno et al presents a novel approach to normal tissue complication modelling using mechanistic modelling. They demonstrate this approach by modelling the effect of fractionation and the impact of radiation quality has on radiation-induced lung fibrosis. They propose this approach as a framework for others to build upon. There are some concerns below which require clarification:

1. In section 3.2 it is mentioned that the ABM-MC model can be fitted by equations (2) and (3) and are qualitatively in agreement with experimental data. It should be clarified what qualitative agreement is and justified that this is sufficient (e.g., are the absolute values from experimental data not reasonable to make a quantitative measurement of agreement?)
2. Last paragraph in section 3.2 – "Overall, the outcomes from the ABM-MC model are in agreement with experimental results found in the literature" should cite what literature is being referred to.
3. Last paragraph in section 3.2 – mention that increased damage of previous model is likely caused by narrower and more peaked dose distributions. Is it not possible to demonstrate if this is the case?
4. Section 3.3 "(from 0.5 to 8.5 Gy)" – Be specific of the doses here. Use a table in the appendix, if need be, just avoid using a range as it doesn't allow reproducibility.
5. Zhou et al mentioned a few times as the experiments matched to for the fractionation work. It would be good to clarify in the text to the reader that these are based on animal experiments.
6. When discussing the effects of fractionation (Figure 4 write-up) please use "total dose" rather than just "dose" to ensure clarity and avoid possible misunderstanding (i.e., it's not fraction dose for the 5#)
7. In the same area there is a comment that FSU survival data for the single fraction deviates from the LQ fit at high doses, it would be good if this was shown in the appendix document.
8. Missing capital letter last sentence of section 3.3
9. Last paragraph section 3.4 RBE_FSU is calculated using the LQ Fit, it is noted that there is a deviation from the LQ Fit at high doses. Whilst I appreciate the wide adoption of the LQ Fit (especially in photons) it is known that it has its limitations (high doses included). Either on the graph or as an appendix graph it would be of interest to include a different fitting function to show the impact of this on RBE. This helps to serve as a warning how sensitive these types of values can be when considering the underlying data rather than just the fit.
10. The discussion has a large amount of repetition from the results section. Please look to reduce this. I believe this section should be more focused on the mechanistic insights (i.e., the benefit of using this more complex approach than an abstract fitting approach).
11. Both the discussion and abstract end by suggesting this work can be used as a framework to investigate RILF. However, there is little information in the Method on how to use said framework. This should be included to fulfil those aspects of the manuscript. It should also serve to improve the reproducibility of the work.
12. AEC1 should be specified on use.
13. It is clear that the method used to come before the results. Please adjust equation numbering, section numbering and first use abbreviations to where they are now first used.
14. Section 2.1 – "Among them, modelling of the apoptotic AEC2s has been implemented to increase the accuracy of the simulations." I didn't see this in the results nor was it discussed in the discussion as far as I saw.
15. Graph captions should make it clear what the error bars represent.
16. Please consider using colour and/or symbols that are clearer, especially for audiences who may be colour blind.

The work discussed in the manuscript very interesting and outlines ABMs as a good tool for modelling in the radiotherapy/radiobiology communities. The coupling to Monte Carlo builds on familiar ground of the community to offer a new approach. With rectification and clarity of the concerns above I believe the manuscript offers value to the research community.

Reviewer #2 (Remarks to the Author):

This manuscript presents an integrated model combining a 3D agent-based biological model with Monte Carlo radiation interaction modelling to predict evolution of a biological system in response to ionising radiation. The authors suggest that this approach effectively reproduces observed biological effects, and can potentially be used as an approach to investigate the mechanistic basis of radiotherapy side effects in the clinic.

I definitely agree with the authors that more integrated biological and physical models have the potential to inform radiotherapy, and the system the authors have put together seems to be largely reasonably designed. However, I have a number of concerns about how its underlying development, and how confident the authors can be that it reflects real biology in a reproducibility quantitative fashion.

One major concern in any such modelling like this is the very large number of parameters - just shy of 100 in this model. With so many parameters, it would be useful to know how many of these actually materially affect model predictions, and how sensitive the model is to these factors. E.g. can some or all of these complex processes be abstracted out entirely without affecting model predictive power? And how certain are the authors in these values? This would provide a lot of insight into the function of the model. Normally I would suggest assessing parameter variance & covariance, but this may be intractable with so many model features and long runtimes. However, some sort of (ideally quantitative) discussion on this point would significantly strengthen the manuscript.

I would also like to see some details about how dose is handled clarified. Specifically, it's unclear from my reading what 'dose' is compared here - e.g. is the ABM treated to a single exact mean dose, and the ABM-MC dose scaled to match the mean dose? Or some other approach? It would be informative to see how the ABM model would perform if it was used with a stochastically sampled dose from a distribution with the same relative standard deviation as a typical MC exposure - would this replicate some or all of the difference in biological effect, without the need for repeated MC calculations?

Also on the point about the radiation dose delivered, 1 keV seems to be a poor choice for the photon exposure. 1 keV photons have an extremely short range in water (~microns). Thus even single cells would significantly shield cells behind them, likely explaining the highly heterogeneous dose distribution the authors observe. In a realistic field (a much higher energy, broad distribution of primarily secondary electrons depositing dose), I would expect the spread in dose across cells to be much narrower, and more in line with the proton data presented in Figure 6. I think this is very significant, as almost all the differences observed between the simulations in subsequent sections seem like they could be explained in large part by a move between a single tight distribution and narrower distributions. Ideally, I'd recommend re-simulating all of this with a realistic clinical treatment field sampled at an appropriate depth. However, I appreciate this may be computationally

infeasible. In the absence of this, I think all of the discussions around effects need to be made with strong caveats that with note that this very broad dose distribution may not represent realistic dose distributions.

This point is particularly significant for the 'RBE' calculations when comparing to proton effects - I don't believe this is a classic 'RBE' you're seeing, but rather just a difference between uniform and heterogeneous dose distributions of a type which isn't typically seen in realistic exposures.

Lastly, the authors in several points note that results are 'in agreement' with other data (or similar phrasing), but in many cases it's ambiguous what data is being referred to, as multiple papers are referred to together. It would significantly strengthen the paper throughout if the authors could be more explicit about what data is being compared to, and ideally making some comment on how similar these trends are - for example, are they just based on overall form of curves, or can quantitative similarities also be drawn? At first glance (assuming I'm looking at the correct datasets) quantitative agreement appears very limited, but it's difficult to make definitive statements about that.

I also have a number of minor comments on presentation and clarification:

- Abstract, second last sentence: When reading this at first, I had the opposite interpretation as intended in the paper (reading 'peaked' and 'flatter' in terms of the spatial distribution, not the dose histogram). I'd suggest "Finally, the model showed increased sensitivity to more uniform proton dose distributions with respect to more heterogeneous ones from photon irradiation" or similar.

- Section and equation numbering seems to be mismatched throughout, maybe due to a formatting issue.

- Figure 2: On the x-axes here, if values are to be in % they should all be scaled by 100; otherwise axis label should read "Dose/DMax" or similar.

- Figure 3-5: IT would be useful to include uncertainty bands on the fir curves, if relevant. In some cases (e.g. Figure 3) it's unclear if some are fully at plateaus, which may lead to quite a bit of uncertainty in the final level.

- Section 3.2, page 4: Here and throughout, it would be good to clarify exactly what is claimed about the level of agreement between models and experimental data, and specifically which data is being referred to.

- Section 3.2, final line: The note about narrower dose distributions here is very relevant, and it would be good if the authors could explicitly test that (e.g. manually varying dose distribution broadness), as it significantly affects the interpretation of the models.

- Section 3.3, final paragraph & figure 5: The authors note that responses couldn't be fit using the sigmoid equations, but this doesn't seem clear - looking at the data, it looks like only the first point would significantly diverge from a sigmoid curve for 5a and 5c? It would be good to check that and see if 'normal' curves are recovered above this low-dose threshold.

- Section 3.3: The authors here considered the impact of modifying only a couple of bystander model parameters - however there numerous other parameters which could also be varied to compensate for these differences. Could they investigate this, or at least comment on the possibility that these

differences may be tied up with other factors (E.g. LQ model parameters).

- Section 3.4: It is alluded to elsewhere, but I think the authors need to be very clear that this isn't an 'RBE' in the traditional sense, as it also reflects a highly different micron-scale dose distribution of a kind which is not typically seen in realistic radiation exposures.

- Discussion, page 9, second paragraph: Here, again, the authors note that "the ABM-MC model preserve the ability to simulate existing experimental results" - however, this is not really strongly quantified, beyond capturing approximate trends, and it would be good to see some more detail on that. Likewise, a degree more in-depth discussion in the following section, possibly by manually varying dose SD in the ABM model, would be useful.

- Discussion, page 10, second paragraph: Here, where RBE is discussed, I think this needs to have a strong caveat that all the RBE effects here may result from the highly heterogeneous dose distribution arising from the 1 keV exposure, and may not be seen in 'real' exposures.

- Methods, agent-based model: Am I correct in reading that the model does not simulate radiation-induced immune cell killing? This is believed to be a significant factor in many aspects of response, and it would be good to describe this in a bit more detail.

- Methods, equation 1: This equation needs some clarification. As written, I can't resolve the quoted values with the figures plotted in the table - I suspect it may be due to how these terms are defined. As written this looks like the absolute increase in ECM against the absolute decrease in surviving FSU volume, but that doesn't seem to match what's plotted. Could this be more clearly defined in the text? Also, this expression does not appear to be dimensionless as may be expected, carrying a unit of $\sqrt{\text{mass}}$ - could the correct units also be noted here?

- Methods, equation 2 and 3: Why are two different sigmoids used here? The two curves under consideration are nearly identical (differing by <0.03 when used with the same parameters, negligibly so when parameter uncertainty is taken into account). It would significantly simplify the discussion if one sigmoid was picked for simplicity, especially as both take the same fitting parameters.

- Methods, equation 3: At first glance it looks like this is saying ΔECM and RSI have the same values, it may be worth sub-scripting or noting in text to indicate that fitting parameters are different for each scenario.

- Methods, page 11, final paragraph on: It would be useful if the authors could clarify why these various different extracellular agents are simulated, and what they add to the model - as noted in the more general comments, can they be abstracted out without affecting the overall model performance?

- Methods, page 11, second paragraph: A feature is noted here where only a portion of AEC2 cells can be depleted. This doesn't seem to be used elsewhere in the paper, that I can see - if it is, this should be explicitly identified, if not it would be best to remove this.

- Methods, Figure 8 and 9: These plots are probably a bit redundant, and could be condensed. Also, it would be useful to include a scale bar for ease of visualization.

- Methods, section 2.3: It would be useful here if the authors would explicitly note how long different runs were performed for (e.g. $N_{\text{steps_ABM}}$? In terms of both simulation steps and simulated time) and provide some detail on statistical sampling, in terms of the number of independent runs performed at each condition and how these were seeded, both within and between runs, to ensure

independence.

- Supplementary material: The ZIP archive with the model code doesn't seem to be available on the reviewer platform, it would be good to confirm this is present for the final submission.

Response to the referees' comments

Reviewer 1

The work detailed in the manuscript “Mechanistic model of radiotherapy induced lung fibrosis using coupled 3D Agent-Based and Monte Carlo simulations” by Cogno et al presents a novel approach to normal tissue complication modelling using mechanistic modelling. They demonstrate this approach by modelling the effect of fractionation and the impact of radiation quality has on radiation-induced lung fibrosis. They propose this approach as a framework for others to build upon. There are some concerns below which require clarification:

1. In section 3.2 it is mentioned that the ABM-MC model can be fitted by equations (2) and (3) and are qualitatively in agreement with experimental data. It should be clarified what qualitative agreement is and justified that this is sufficient (e.g., are the absolute values from experimental data not reasonable to make a quantitative measurement of agreement?)

Thank you for pointing this out, indeed being able to replicate experimental results quantitatively would prove the trustworthiness of the model. However, the outcomes that the model can replicate are mainly the increase in the ECM concentration at early and late phases as well as the RSI, which replaces the fibrosis index introduced by Zhou et al. in <https://pubmed.ncbi.nlm.nih.gov/29116014/>. Both these metrics are written in terms of the variation in the absorbance of the fibrotic lungs which is measured in Hounsfield Units (HU). However, in our model this change is measured in g/cm^3 , as the radiodensity can't be replicated. Moreover, the decrease in the functioning volume used to express the FI lacks a direct correspondence in our model, as its simulation would require much additional complexity. Therefore, the RSI that we introduced is written in terms of the alveolar volume that survives the irradiation (i.e. contains viable AEC2). Overall, these differences make a quantitative comparison not always straightforward, however a qualitative one is achievable. In fact, our model could replicate both the sigmoidal trends of the ΔECM (at early and late phases) and that of the RSI showed in previous studies, as well as the distinctive peak of ΔECM between 3 and 4 months after the irradiation. This has now been clarified in the aforementioned section. A quantitative agreement between the RSI and the FI for photons in 1 fraction (and partly in 5 fractions) was observed. This was shown in the results and mentioned in the discussion.

2. Last paragraph in section 3.2 – “Overall, the outcomes from the ABM-MC model are in agreement with experimental results found in the literature” should cite what literature is being referred to.

Thank you for the comment. The relevant literature has been added both in the mentioned paragraph and throughout the whole section.

3. Last paragraph in section 3.2 – mention that increased damage of previous model is likely caused by narrower and more peaked dose distributions. Is it

not possible to demonstrate if this is the case?

We have increased the photon energy and obtained narrower dose distributions. With that, we re-simulated the RILI at all doses and obtained outcomes closer to those of our old model. These results have been added to section 2.2. Thank you for this constructive suggestion.

4. Section 3.3 “(from 0.5 to 8.5 Gy)” – Be specific of the doses here. Use a table in the appendix, if need be, just avoid using a range as it doesn’t allow reproducibility.

A small table that recapitulates all the simulated doses was added at the end of the paragraph, thank you.

5. Zhou et al mentioned a few times as the experiments matched to for the fractionation work. It would be good to clarify in the text to the reader that these are based on animal experiments.

Thank you, this has now been mentioned both in the results and discussion.

6. When discussing the effects of fractionation (Figure 4 write-up) please use “total dose” rather than just “dose” to ensure clarity and avoid possible misunderstanding (i.e., it’s not fraction dose for the 5#)

This has been made more explicit, thank you.

7. In the same area there is a comment that FSU survival data for the single fraction deviates from the LQ fit at high doses, it would be good if this was shown in the appendix document.

Thank you for pointing this out. In that section we actually referred to the survival probability expressed by equation (5), which was derived from the critical volume NTCP model with LQ parameters. This has now been clarified. Moreover, using the more realistic dose distribution obtained from the 10keV photons we obtained an FSU survival that showed a higher degree of agreement with equation (5) with respect to the older one.

8. Missing capital letter last sentence of section 3.3

Fixed, thank you.

9. Last paragraph section 3.4 RBE_FSU is calculated using the LQ Fit, it is noted that there is a deviation from the LQ Fit at high doses. Whilst I appreciate the wide adoption of the LQ Fit (especially in photons) it is known that it has its limitations (high doses included). Either on the graph or as an appendix graph it would be of interest to include a different fitting function to show the impact of this on RBE. This helps to serve as a warning how sensitive these types of values can be when considering the underlying data rather than just the fit.

Thank you for this input. First of all, we apologize for any potential confusion. What we actually meant in the aforementioned section is that for the fit we used the LQ model adapted to the FSU survival (LQ_{FSU}) which is expressed by equation (5), where also the number of stem cells is taken into account. This has now been clarified. Moreover, by using the narrower dose distribution for the photons that we obtained when their energy was increased, we observed a high level of agreement between the fit curve and the data. Such a high degree of agreement was already observed for the protons, whose dose distribution is more homogeneous than that of the photons. This notwithstanding, we fitted the dataset using equation (5), but we replaced the LQ cell survival curve with the Linear-Quadratic-Linear (LQL_{FSU}) model and with the Universal Survival Curve (USC_{FSU}). The LQL_{FSU} and USC_{FSU} fit parameters that

we obtained were almost identical to those of the LQ_{FSU} and thus no difference in the RBE_{FSU} could be seen. We also tried to use the USC and LQL curves without the FSU adaptation (i.e. as they're normally used for the cell survival), but the data displayed marked deviations from the fitted curves and a prominent shoulder in the dose range around the threshold dose D_T . We have thus included the data for photons and protons and the fitted curves (USC_{FSU} and LQL_{FSU} only) in the supplementary figure S3.

10. The discussion has a large amount of repetition from the results section. Please look to reduce this. I believe this section should be more focused on the mechanistic insights (i.e., the benefit of using this more complex approach than an abstract fitting approach).

Thank you for the suggestion, we have revised the discussion and shortened it by keeping the main results, but removing unnecessary repetitions. Moreover, at the end of it we have presented some of the benefits of using our approach and mentioned that our manuscript shows that it has the potential to inform further radiobiological studies.

11. Both the discussion and abstract end by suggesting this work can be used as a framework to investigate RILF. However, there is little information in the Method on how to use said framework. This should be included to fulfil those aspects of the manuscript. It should also serve to improve the reproducibility of the work.

Thank you for pointing this out, a description of the required steps has been added as a paragraph at the end of the Methods section.

12. AEC1 should be specified on use.

Added, thank you.

13. It is clear that the method used to come before the results. Please adjust equation numbering, section numbering and first use abbreviations to where they are now first used.

Adjusted, thank you.

14. Section 2.1 – “Among them, modelling of the apoptotic AEC2s has been implemented to increase the accuracy of the simulations.” I didn't see this in the results nor was it discussed in the discussion as far as I saw. Thank you for mentioning this. In the old model the AEC2s that were killed by the radiation were removed right away. The new model accounts for their presence in the alveolar space by providing the apoptotic state. Their number is low throughout the simulation as the removal time is relatively short and the trend can be seen in the plots provided in the supplementary material.

15. Graph captions should make it clear what the error bars represent.

Thank you for the comment, these were added in all the captions.

16. Please consider using color and/or symbols that are clearer, especially for audiences who may be color blind.

Thank you for pointing this out. This aspect was overlooked by us, but has now been fixed by replacing the old colors in the plots with new palette.

The work discussed in the manuscript is very interesting and outlines ABMs as a good tool for modelling in the radiotherapy/radiobiology communities. The coupling to Monte Carlo builds on familiar ground of the community to offer a new approach. With rectification and clarity of the concerns above I believe the manuscript offers value to the research community.

Reviewer 2

This manuscript presents an integrated model combining a 3D agent-based biological model with Monte Carlo radiation interaction modelling to predict evolution of a biological system in response to ionising radiation. The authors suggest that this approach effectively reproduces observed biological effects, and can potentially be used as an approach to investigate the mechanistic basis of radiotherapy side effects in the clinic.

I definitely agree with the authors that more integrated biological and physical models have the potential to inform radiotherapy, and the system the authors have put together seems to be largely reasonably designed. However, I have a number of concerns about how its underlying development, and how confident the authors can be that it reflects real biology in a reproducibility quantitative fashion.

- One major concern in any such modelling like this is the very large number of parameters - just shy of 100 in this model. With so many parameters, it would be useful to know how many of these actually materially affect model predictions, and how sensitive the model is to these factors. E.g. can some or all of these complex processes be abstracted out entirely without affecting model predictive power? And how certain are the authors in these values? This would provide a lot of insight into the function of the model. Normally I would suggest assessing parameter variance & covariance, but this may be intractable with so many model features and long runtimes. However, some sort of (ideally quantitative) discussion on this point would significantly strengthen the manuscript.

Thank you for this insightful comment. As you mentioned, the model features almost 100 parameters that can be roughly categorized as follows: cell turnover parameters, reaction-diffusion parameters, damage-associated parameters, geometry-related parameters. The cell turnover parameters are based on an established model of idiopathic pulmonary fibrosis and tuned so that our model could both replicate the results of this study and maintain homeostatic conditions if not damaged. The geometry-related and reaction-diffusion parameters were mostly found in the literature from experimental studies. Lastly, due to the limited amount of available data, the damage-associated parameters have been estimated indirectly, i.e. tuned so that the model's predictions would be clinically plausible.

We performed a partial sensitivity analysis in our first ABM which indicated the most influential parameters both on the ECM concentration and overall. The parameters associated with the mechanisms of the radiation-induced damage were then determined in our second ABM, such that the trends observed in experimental results could be replicated by our model. However, only the effect of the damaged/senescent ratio was thoroughly analyzed. This notwithstanding, as you pointed out a full sensitivity analysis with such long runtimes and high number of features would be intractable. To overcome this deficiency, we have now extended the section "Impact of the bystander threshold" (renamed as "Targeted response assessment"), where we compared the effects of altering a few parameters (including the bystander threshold) on the model's outcomes and commented on the extent of these impacts.

- I would also like to see some details about how dose is handled clarified. Specifically, it's unclear from my reading what 'dose' is compared here - e.g. is the ABM treated to a single exact mean dose, and the ABM-MC dose scaled to match the mean dose? Or some other approach? It would be informative to see how the ABM model would

perform if it was used with a stochastically sampled dose from a distribution with the same relative standard deviation as a typical MC exposure - would this replicate some or all of the difference in biological effect, without the need for repeated MC calculations?

Thank you for pointing this out, this is indeed not clear. The MC simulation provides an absorbed dose for each cell. This dose is very low and assuming a constant LET along the particles' tracks (an assumption explicitly stated in the manuscript) the dose can be considered dependent only on the fluence, and hence the number of histories. We've selected a number of histories high enough such that the differences between the average doses per alveolus are small (given our configuration with 4 external fields), but the delivered dose per cell is smaller than the minimum dose that we wanted to simulate. After the irradiation phase, an average dose per cell is computed in the ABM and scaled to the desired dose. Finally, this scaled dose is used to compute the survival probability for each cell using the LQ model and following a comparison against a random number the cell's fate determined. This has now been clarified in subsection 4.2, "Coupling the models". As for using a stochastically sampled dose from a distribution as a typical MC exposure, this is unfortunately not feasible at the current stage, as the dose deposition is simulated ad hoc for the custom alveolar segment (whose structure was rebuilt in TOPAS-nBio) and our model aims at replicating cell scale radiobiological mechanisms. As such, our goal was to investigate the possibility to simulate long term radiation damage given a "per-cell" dose distribution instead of the more traditional dose per voxel. This notwithstanding, following your constructive comments we have performed simulations with a higher photon energy (please, see the reply to the next comment) and observed a decrease in the standard deviation of the exposure. This has been used to run the ABM and indeed we observed differences in the biological effect.

- Also on the point about the radiation dose delivered, 1 keV seems to be a poor choice for the photon exposure. 1 keV photons have an extremely short range in water (~microns). Thus even single cells would significantly shield cells behind them, likely explaining the highly heterogeneous dose distribution the authors observe. In a realistic field (a much higher energy, broad distribution of primarily secondary electrons depositing dose), I would expect the spread in dose across cells to be much narrower, and more in line with the proton data presented in Figure 6. I think this is very significant, as almost all the differences observed between the simulations in subsequent sections seem like they could be explained in large part by a move between a single tight distribution and narrower distributions. Ideally, I'd recommend re-simulating all of this with a realistic clinical treatment field sampled at an appropriate depth. However, I appreciate this may be computationally infeasible. In the absence of this, I think all of the discussions around effects need to be made with strong caveats that with note that this very broad dose distribution may not represent realistic dose distributions.

This point is particularly significant for the 'RBE' calculations when comparing to proton effects - I don't believe this is a classic 'RBE' you're seeing, but rather just a difference between uniform and heterogeneous dose distributions of a type which isn't typically seen in realistic exposures.

Thank you for this comment, indeed increasing the energy of the photons narrowed the dose distribution.

As mentioned before, our model, at the current stage, is aimed at simulating dose depositions at the cell scale. Therefore, using a per-voxel dose distribution from a clinical treatment field would be out of scope and unfeasible. However, that's definitely something we could look into when moving to a larger scale, thank you.

This notwithstanding, we re-simulated the effect of all the doses (and multi-fraction regimes) on the radiation-induced injury at higher photon energies and observed outcomes closer to our previous ABM model and to those obtained from the proton irradiation. Moreover, the RBE (whose actual interpretation has been stressed more in the "Results" and "Discussion" sections) has consequently changed and has become smaller. A strong caveat has therefore been added about the dose distributions and the new results have been added to the "Results" section.

Multiple photon energies have been tested, from 10keV to 6MeV. The original energy, 1keV, was chosen so as to have a high cross section and thus lower the number of histories needed (and the computational power). With the higher energies, we've increased also the number of histories. However, the dose distributions associated to $E_\gamma > 10\text{keV}$ appeared either multi-peaked or with a high number of counts at 0 Gy, possibly due to very low cross sections. Despite the small increase in the photon energy employed for the new simulations (from 1 to 10 keV), the resulting dose distribution, centered on the mean value, has narrowed substantially and resembles the protons one. It's also worth noting that the volume of the simulated cubic space is very small (8 mm^3) and the distance between the source and the alveolar segment in the MC simulator is a few hundred micrometers.

- Lastly, the authors in several points note that results are 'in agreement' with other data (or similar phrasing), but in many cases it's ambiguous what data is being referred to, as multiple papers are referred to together. It would significantly strengthen the paper throughout if the authors could be more explicit about what data is being compared to, and ideally making some comment on how similar these trends are - for example, are they just based on overall form of curves, or can quantitative similarities also be drawn? At first glance (assuming I'm looking at the correct datasets) quantitative agreement appears very limited, but it's difficult to make definitive statements about that.

This has been clarified more and revised, thank you. Indeed the comparisons in the manuscript were ambiguous and we have addressed this issue by specifying more precisely the experimental data from other studies we referred to. You are correct in that the quantitative agreement is limited, and in the manuscript we showed that our model can qualitatively replicate trends from clinical and laboratory data (such as those of the ECM and the RILI severity index). However, in addition to those, our model could replicate other qualitative features that have been observed in the clinical settings, such as patterns of early and late effects (seen in the ECM) where the time of the maximum ECM concentration matched that of clinical results (~ 3 months, as shown in the Supplementary Material), complete recovery for low doses (as observed in previous experiments) and increased survival (and overall outcome) for the same total dose when exposed to a 5-fraction scheme with respect to a single fraction. The level of agreement could not be quantified for these features, but nevertheless we highlighted the ability of the model to match these observations despite the low availability of experimental data. A quantitative agreement between the RSI and the FI for photons in 1 fraction (and partly in 5 fractions) was observed. This was shown in the results and mentioned in the discussion.

I also have a number of minor comments on presentation and clarification:

1. Abstract, second last sentence: When reading this at first, I had the opposite interpretation as intended in the paper (reading 'peaked' and 'flatter' in terms of the spatial distribution, not the dose histogram). I'd suggest "Finally, the model showed increased sensitivity to more uniform proton dose distributions with respect to more heterogeneous ones from photon irradiation" or similar.

Thank you for pointing that out, I was referring to the dose histograms as you suggested. I've changed the sentence.

2. Section and equation numbering seems to be mismatched throughout, maybe due to a formatting issue.

Fixed, thank you.

3. Figure 2: On the x-axes here, if values are to be in % they should all be scaled by 100; otherwise axis label should read "Dose/DMax" or similar.

I've replaced the label with "Dose/Dose_{max}", thank you.

4. Figure 3-5: IT would be useful to include uncertainty bands on the fit curves, if relevant. In some cases (e.g. Figure 3) it's unclear if some are fully at plateaus, which may lead to quite a bit of uncertainty in the final level.

Thank you for this input. We've computed the uncertainty bands, but we've decided to avoid plotting them as they're quite small. However, we have improved the visibility of our plots and clarified how the errors have been computed.

5. Section 3.2, page 4: Here and throughout, it would be good to clarify exactly what is claimed about the level of agreement between models and experimental data, and specifically which data is being referred to.

Thank you for the comment, claims about the level of agreement between models and experimental data have been clarified throughout the text.

6. Section 3.2, final line: The note about narrower dose distributions here is very relevant, and it would be good if the authors could explicitly test that (e.g. manually varying dose distribution broadness), as it significantly affects the interpretation of the models.

This has been tested and all the plots updated as increasing the photon energy (as suggested by you) led to narrower and thus more realistic dose distributions which, in turn, shifted the results towards those of the old model. The improvements are now shown in figures 2 and 3. Thank you very much for this constructive comment.

7. Section 3.3, final paragraph & figure 5: The authors note that responses couldn't be fit using the sigmoid equations, but this doesn't seem clear - looking at the data, it looks like only the first point would significantly diverge from a sigmoid curve for 5a and 5c? It would be good to check that and see if 'normal' curves are recovered above this low-dose threshold.

This has now been fixed thanks to the new narrower dose distribution. Thank you for pointing it out.

8. Section 3.3: The authors here considered the impact of modifying only a couple of bystander model parameters - however there are numerous other parameters which could also be varied to compensate for these differences. Could they investigate this, or at least comment on the possibility that these differences may be tied up with other factors (E.g. LQ model parameters).

Thank you for this input. We investigated the effect of small variations in two

parameters (namely, the cells radiosensitivity and the damaged-to-senescent rate) to see whether their reduction could compensate the effect of the bystander parameters. We renamed the section “Impact of the bystander threshold” to “Targeted response assessment” where we compared how the parameters’ changes affected the model’s predictions. Moreover, we removed the effect of the parameters on the multi-fractionation scenario as they didn’t add anything new to the results.

9. Section 3.4: It is alluded to elsewhere, but I think the authors need to be very clear that this isn't an 'RBE' in the traditional sense, as it also reflects a highly different micron-scale dose distribution of a kind which is not typically seen in realistic radiation exposures.

This has been clarified more, thank you.

10. Discussion, page 9, second paragraph: Here, again, the authors note that "the ABM-MC model preserve the ability to simulate existing experimental results" - however, this is not really strongly quantified, beyond capturing approximate trends, and it would be good to see some more detail on that. Likewise, a degree more in-depth discussion in the following section, possibly by manually varying dose SD in the ABM model, would be useful.

Thank you for this observation. As mentioned in the answers to the general comments, this has been addressed by adding more details about the experimental data we referred to when making comparisons throughout the whole manuscript. As for the dose SD, we couldn’t manually change it as the dose distribution resulted from the absorbed dose in the alveolar segment structure as computed by the MC. Following the MC simulation, a text file was generated that associated each cell (given its ID) to a certain dose. In fact, our model aimed at measuring the dose deposition at the cell scale as an input for the ABM, rather than the dose per-voxel. This notwithstanding, we actually reduced the dose SD by increasing the energy of the photons, as suggested by you. The quantitative agreement observed between the RSI and the FI for photons in 1 fraction was added to the discussion.

11. Discussion, page 10, second paragraph: Here, where RBE is discussed, I think this needs to have a strong caveat that all the RBE effects here may result from the highly heterogeneous dose distribution arising from the 1 keV exposure, and may not be seen in 'real' exposures.

Thank you for this important note. It has now been highlighted at the end of the aforementioned paragraph and in the discussion.

12. Methods, agent-based model: Am I correct in reading that the model does not simulate radiation-induced immune cell killing? This is believed to be a significant factor in many aspects of response, and it would be good to describe this in a bit more detail.

Thank you for the note. Yes, indeed the radiation-induced immune cell killing was not simulated. Although we know about the importance of the factor, we decided not to take that into account to avoid additional complexity and uncertainty in the model parameters. However, the model features two parameters that allow the user to set the number of senescence cells that each macrophage can clear as well as the fraction of macrophages that are active and can successfully phagocyte the senescent cells. Users can thus combine these two parameters to simulate a reduction in the capabilities of the immune system caused by a radiation-induced damage. Moreover, it was mentioned in the manuscript that these two parameters were investigated as they heavily affected the simulation results. This notwithstanding, thank you for

highlighting this deficiency. We added a paragraph in the Methods section where we pointed it out and mentioned possible steps to extend the model to simulate the immune cell killing.

13. **Methods, equation 1: This equation needs some clarification. As written, I can't resolve the quoted values with the figures plotted in the table - I suspect it may be due to how these terms are defined. As written this looks like the absolute increase in ECM against the absolute decrease in surviving FSU volume, but that doesn't seem to match what's plotted. Could this be more clearly defined in the text? Also, this expression does not appear to be dimensionless as may be expected, carrying a unit of $\sqrt{\text{mass}}$ - could the correct units also be noted here?**

Thank you for the comment. As you mentioned, the RSI is given by the absolute increase in ECM against the absolute decrease in surviving FSU volume, but this wasn't clearly expressed in the manuscript. A more detailed explanation about the RSI calculation has been added in the "Methods" section as well as the correct unit. All the plots showing the RSI have been redrawn with the correct unit and scale factor.

14. **Methods, equation 2 and 3: Why are two different sigmoids used here? The two curves under consideration are nearly identical (differing by <0.03 when used with the same parameters, negligibly so when parameter uncertainty is taken into account). It would significantly simplify the discussion if one sigmoid was picked for simplicity, especially as both take the same fitting parameters.**

The two definitions were introduced to distinguish between the early and the late components of the ECM increase. However, as pointed out, the curves are nearly identical and might make the discussion more confusing. The first one (equation (3), was (2) before the update) has now been used for the delta ECM, so as to have equation (4) ((3) in the first draft) only for the RSI. Everything has been re-plotted accordingly. Thank you for the suggestion.

15. **Methods, equation 3: At first glance it looks like this is saying ΔECM and RSI have the same values, it may be worth sub-scripting or noting in text to indicate that fitting parameters are different for each scenario.**

Thank you for the comment. Following your previous suggestion equation (4) (was (3)) is not used for the Delta ECM anymore and thus the note is not necessary anymore. On a side note, a "(D)" was added in the left-hand side of the equations (3), (4) and (5) to make the dependence on the dose more explicit.

16. **Methods, page 11, final paragraph on: It would be useful if the authors could clarify why these various different extracellular agents are simulated, and what they add to the model - as noted in the more general comments, can they be abstracted out without affecting the overall model performance?**

Thank you for the comment. Diffusion, decay and, in some cases, depletion, of multiple substances have been included in the model since its first implementation not only to provide a more realistic representation of the underlying network of interactions, but also to make it more customizable by allowing users to simulate the effect of potential drugs/therapies. Moreover, the differences in the roles of the extracellular substances make it difficult to abstract them and simplify the network. In fact, 10 substances are simulated in total (their time evolution can be found in the Supplementary Material) which act as an interface between different cell populations. The role of the different substances was documented in Cogno et al. 2022 (<https://www.mdpi.com/2073-8994/14/1/90>) and Cogno et al. 2022a (<https://www.mdpi.com/1422-0067/23/22/13920>) and thus it was excluded from this work to avoid redundancy.

Among the substances, for example, the MCP-1 is secreted initially by the senescent AEC2 to gather the monocytes, while the macrophages act on the mesenchymal cells' proliferation by secreting TGFbeta. Besides, the macrophages can adjust the ECM concentration by secreting TIMP and MMP, which are mentioned in the manuscript to highlight the employment of the newly-implemented substance depletion feature. This notwithstanding, this wasn't clarified in the manuscript and has now been added to the Methods (end of the mentioned paragraph). Thank you for bringing this to our attention.

17. Methods, page 11, second paragraph: A feature is noted here where only a portion of AEC2 cells can be depleted. This doesn't seem to be used elsewhere in the paper, that I can see - if it is, this should be explicitly identified, if not it would be best to remove this.

Thank you for pointing this out. Indeed the mentioned feature wasn't employed in the simulations presented in the manuscript. The paragraph has been removed.

18. Methods, Figure 8 and 9: These plots are probably a bit redundant, and could be condensed. Also, it would be useful to include a scale bar for ease of visualization. The figures were merged and a scale bar added, thank you for the suggestion.

19. Methods, section 2.3: It would be useful here if the authors would explicitly note how long different runs were performed for (e.g. N_steps_ABM? In terms of both simulation steps and simulated time) and provide some detail on statistical sampling, in terms of the number of independent runs performed at each condition and how these were seeded, both within and between runs, to ensure independence.

Thank you for the comment, these are indeed crucial information and have been added to the subsection "Coupling the models".

20. Supplementary material: The ZIP archive with the model code doesn't seem to be available on the reviewer platform, it would be good to confirm this is present for the final submission.

This is indeed indispensable and was included in the first submission, we apologize for its unavailability. We'll make sure it's available in the final submission, thank you.

REVIEWERS' COMMENTS:

Reviewer #1 (Remarks to the Author):

The authors have diligently answered my concerns from my initial review. I am satisfied by the modifications made. These additions have made the manuscript stronger. I now believe it offers good value to the research community as a novel approach to modelling lung fibrosis and recommend it for publication in this journal.

Reviewer #2 (Remarks to the Author):

I'd like to thank the authors for their thorough response to my questions, I think the updates have improved the manuscript significantly. Bearing in mind the challenges of fully validating these models, I'm happy with the updates the authors have made, and to recommend it proceed to publication.